# On the Interaction of Compressibility and Adversarial Robustness

## Abstract

Modern neural networks are expected to simultaneously satisfy a host of desirable properties: accurate fitting to training data, generalization to unseen inputs, parameter and computational efficiency, and robustness to adversarial perturbations. While compressibility and robustness have each been studied extensively, a unified understanding of their interaction still remains elusive. In this work, we develop a principled framework to analyze how different forms of compressibility - such as neuron-level sparsity and spectral compressibility - affect adversarial robustness. We show that these forms of compression can induce a small number of highly sensitive directions in the representation space, which adversaries can exploit to construct effective perturbations. Our analysis yields a simple yet instructive robustness bound, revealing how neuron and spectral compressibility impact $\ell_\infty$ and $\ell_2$ robustness via their effects on the learned representations. Crucially, the vulnerabilities we identify arise irrespective of how compression is achieved - whether via regularization, architectural bias, or implicit learning dynamics. Through empirical evaluations across synthetic and realistic tasks, we confirm our theoretical predictions, and further demonstrate that these vulnerabilities persist under adversarial training and transfer learning, and contribute to the emergence of universal adversarial perturbations. Our findings show a fundamental tension between structured compressibility and robustness, and suggest new pathways for designing models that are both efficient and secure.

## 1 Introduction

Machine learning (ML) systems are increasingly deployed in high-stakes domains such as healthcare (Rajpurkar et al., 2022) and autonomous driving (Hussain & Zeadally, 2019), where reliability is paramount. With their growing social impact, modern neural networks are now expected to meet a suite of often conflicting demands: they must **fit the data** (explain observations), **generalize** to unseen inputs, remain efficient in storage and inference, *i.e.*, be **compressible**, and exhibit **robustness** against adversarial perturbations, as well as other distribution shifts. While each of these desiderata has been studied extensively in isolation, a mature and unified understanding of how they interact—and in particular, how compressibility shapes robustness—remains elusive.

As desirable as adversarial robustness and compressibility both are, the research has been equivocal regarding whether/when/how their simultaneous achievement is possible (Guo et al., 2018; Balda et al., 2020; Li et al., 2020a; Merkle et al., 2022; Liao et al., 2022; Piras et al., 2024). However, recent work has started to provide mechanism-based explanations for the relationship between the two, highlighting how compressibility impacts models' vulnerability to adversarial noise. For example, Savostianova et al. (2023) demonstrate that low-rank parameterizations may inadvertently amplify local Lipschitz constants, increasing sensitivity to perturbations. Nern et al. (2023) connect adversarial transferability to layer-wise operator norms and their impact on representation geometry.

Submitted to 39th Conference on Neural Information Processing Systems (NeurIPS 2025). Do not distribute.

Feng et al. (2025) further shows that while moderate sparsity can enhance robustness, excessive sparsity causes ill-conditioning that reintroduces fragility and vulnerability. These results hint at a delicate, regime-dependent relationship between compressibility and robustness—but a principled and general framework is still lacking.

In this work, we develop a framework to investigate the effect of structured sparsity on adversarial robustness through its effect on parameter operator norms and network's Lipschitz constant. We jointly study how different forms of compressibility—particularly neuron-level sparsity and spectral compression—affect adversarial robustness. Our central result is an intuitive and instructive adversarial robustness bound that reveals how compressibility can induce a small set of highly sensitive directions in the representation space. These "adversarial axes" dramatically amplify perturbations and are readily exploited by adversaries. Empirically, we confirm that these axes are not merely theoretical constructs: adversarial attacks reliably identify and exploit them across architectures, datasets, and attack models. Previous research tightly links compressibility to generalization (Arora et al., 2018; Barsbey et al., 2021); however, our findings imply that the very mechanisms that promote generalization can also introduce structural weaknesses. In summary, our contributions are:

1. We provide an adversarial robustness bound that decomposes into analytically interpretable terms, and predicts that neuron and spectral compressibility create adversarial vulnerability against $\ell_\infty$ and $\ell_2$ attacks, through their effects on networks' Lipschitz constants.
2. Utilizing various compressibility-inducing interventions, we empirically validate our predictions regarding the emergence of adversarial vulnerability under structured compressibility.
3. We demonstrate that the detrimental effects of compressibility persist under adversarial training and transfer learning, and can contribute to the appearance of universal adversarial examples.
4. We demonstrate that the compressed models inherit the negative effects of compressibility, and leverage our bound to propose regularization and pruning strategies that are simple yet effective.

We will make our implementation publicly available upon publication.

## 2 Setup

**Notation**. We denote scalars by lower case italic ($k$), vectors with lower case bold ($\boldsymbol{x}$), and matrices with upper case bold ($\mathbf{W}$) characters respectively. Vector $\ell_p$ norms are denoted by $\|\boldsymbol{x}\|_p$. For matrices, $\|\mathbf{W}\|_F, \|\mathbf{W}\|_2, \|\mathbf{W}\|_\infty$ correspond to Frobenius, spectral, and $\ell_\infty$ operator norms, respectively. We denote the $i^{\text{th}}$ element of a vector $\boldsymbol{x}$ with $x_i$, and row $i$ of a matrix $\mathbf{W}$ with $\mathbf{w}_i$. Elements of a sequence of matrices (e.g. layer matrices) are referred to by $\mathbf{W}^l, l \in [\lambda]$. For an integer $n$, we use $[n] := (1, \ldots, n)$.

Unless otherwise specified, we will be focusing on supervised classification problems, which will involve the input $\boldsymbol{x} \in \mathcal{X}$ and label $y \in \mathcal{Y}$. A predictor $g : \mathcal{X} \to \mathbb{R}^{|\mathcal{Y}|}$, parametrized by $\boldsymbol{\theta} \in \Theta$ produces output logits $\boldsymbol{s} = g(\boldsymbol{x}, \boldsymbol{\theta})$, the maximum of which is the predicted label $\hat{y} = \arg\max_{i \in |\mathcal{Y}|} s_i$. Predictions are evaluted by a loss function $\ell : \mathbb{R}^{|\mathcal{Y}|} \times \mathcal{Y} \to \mathbb{R}_+$. For brevity, we define the composite loss function $f(\boldsymbol{x}, \boldsymbol{\theta}) := \ell(g(\boldsymbol{x}, \boldsymbol{\theta}), y)$.

**Risk and adversarial robustness**. Assuming a data distribution $\pi$ on $\mathcal{X} \times \mathcal{Y}$, we define the population and empirical risks accordingly: $F(\boldsymbol{\theta}) := \mathbb{E}_{\boldsymbol{x}, y \sim \pi}[f(\boldsymbol{x}, \boldsymbol{\theta})]$, and $\widehat{F}(\boldsymbol{\theta}, S) := \frac{1}{n} \sum_{i=1}^{n} f(\boldsymbol{x}_i, \boldsymbol{\theta})$, where $(\boldsymbol{x}_i, y_i)_{i=1}^n$ denotes a set of i.i.d. samples from $\pi$. Adversarial attacks are minimal perturbations to input that dramatically disrupt a model's predictions (Szegedy et al., 2014). In this paper, we focus on bounded $p$-norm attacks, which we define as

$$\boldsymbol{a}^* = \underset{\|\boldsymbol{a}\|_p \leq \delta}{\arg\max} f(\boldsymbol{x} + \boldsymbol{a}, \boldsymbol{\theta}). \tag{1}$$

Given the adversarial loss $f_p^{\text{adv}}(\boldsymbol{x}, \boldsymbol{\theta}; \delta) := f(\boldsymbol{x} + \boldsymbol{a}^*, \boldsymbol{\theta})$, we define adversarial risk and empirical adversarial risk as $F_p^{\text{adv}}(\boldsymbol{\theta}; \delta) := \mathbb{E}_{\boldsymbol{x} \sim \pi}[f_p^{\text{adv}}(\boldsymbol{x}, \boldsymbol{\theta}; \delta)]$ and $\widehat{F}_p^{\text{adv}}(\boldsymbol{\theta}, S; \delta) := \frac{1}{n} \sum_{i=1}^{n} f_p^{\text{adv}}(\boldsymbol{x}_i, \boldsymbol{\theta}; \delta)$, respectively. The type of the selected *attack norm* $p$ for the *attack budget* $\delta$, determines the type of adversarial attack in question, with $p = 2$ and $p = \infty$ as the most common choices. In this paper, we are primarily interested in what we call the *adversarial robustness gap*: $\Delta_p^{\text{adv}} := F_p^{\text{adv}}(\boldsymbol{\theta}, \delta) - F(\boldsymbol{\theta})$. A model with small $\Delta_p^{\text{adv}}$ is considered *adversarially robust*.

**Neural networks**. Our analyses will focus on neural networks under classification. We define a fully connected neural network (FCN) with $\lambda$ hidden layers of $h$ units as below:

$$g(\boldsymbol{x}, \boldsymbol{\theta}) = \mathbf{C}\phi(\mathbf{W}^{\lambda}\phi(\dots \mathbf{W}^{1}\boldsymbol{x})), \tag{2}$$

where $\boldsymbol{\theta} := (\mathbf{C}, \mathbf{W}^{1}, \dots, \mathbf{W}^{\lambda})$, $\phi$ is elementwise ReLU activation function. We can write $g$ as the composition of two functions, a linear classifier head $c : \mathbb{R}^{h} \to \mathbb{R}^{|\mathcal{Y}|}$, and a feature encoder $\Phi : \mathcal{X} \to \mathbb{R}^{h}$, such that $g(\boldsymbol{x}, \boldsymbol{\theta}) := c(\cdot, \mathbf{C}) \circ \Phi(\cdot, \mathbf{W}^{1} \dots \mathbf{W}^{\lambda})(\boldsymbol{x})$. To avoid notational clutter and without loss of generality, throughout our analyses we assume that $\boldsymbol{x} \in \mathbb{R}^{h}$, and omit bias parameters.

**Lipschitz continuity**. Given two $L^{p}$ spaces $\mathcal{X}$ and $\mathcal{Y}$, a function $g : \mathcal{X} \to \mathcal{Y}$ is called Lipschitz continuous if there exists a constant $K_{p}$ such that $\|g(\boldsymbol{x}^{1}) - g(\boldsymbol{x}^{2})\|_{p} \leq K_{p}\|\boldsymbol{x}^{1} - \boldsymbol{x}^{2}\|_{p}, \forall \boldsymbol{x}^{1}, \boldsymbol{x}^{2} \in \mathcal{X}$. Said $K_{p}$ is called the (global) Lipschitz constant. Any $\bar{K}_{p}$ that is valid for a subspace $\mathcal{U} \subset \mathcal{X}$ is called a local Lipschitz constant. Although its computation is NP-hard for even the simplest neural networks (Scaman & Virmaux, 2018); as a notion of input-based volatility, estimation, utilization, and regularization of the Lipschitz constant have been a staple of robustness research (Cisse et al., 2017; Bubeck et al., 2020; Grishina et al., 2025). Note that the FCN as defined in Eq (2) is Lipschitz continuous in $\ell_{p}$ for $p \in [1, \infty]$, along with other commonly used architectures such as convolutional neural networks (CNN) (Zühlke & Kudenko, 2025).

**Compressibility**. Various prominent approaches to neural network compression exist, such as pruning, quantization, distillation, and conditional computing, (O'Neill, 2020). Here we focus on pruning, which is arguably the most commonly used and researched form of compression (Hohman et al., 2024). More specifically, we focus on inherent properties of network parameters that make them amenable to pruning, *i.e.* their *compressibility*. We continue with a formal definition.

**Definition 2.1** (($q, k, \epsilon$)-compressibility)**.** *Given a vector $\boldsymbol{\theta} \in \mathbb{R}^{d}$ and a non-negative integer $k \leq d$, let $\boldsymbol{\theta}_{k}$ denote the compressed vector which contains the largest (in magnitude) $k$ elements of $\boldsymbol{\theta}$ with all the other elements set to $0$. Then, $\boldsymbol{\theta}$ is ($q, k, \epsilon$)-compressible if and only if*

$$\|\boldsymbol{\theta} - \boldsymbol{\theta}_{k}\|_{q} / \|\boldsymbol{\theta}\|_{q} \leq \epsilon. \tag{3}$$

*In the case of equality, we call $\boldsymbol{\theta}$ to be strictly ($q, k, \epsilon$)-compressible.*

Moving forward we will assume any vector denoted as compressible is strictly compressible, unless otherwise noted. The concept of compressibility can be thought of as the generalization of *sparsity*, with the obvious advantage of being applicable to domains where true sparsity is rare, such as neural network parameter values. Note that our intuitive definition of compressibility is based on foundational results in compressed sensing and is well exploited in the established machine learning literature (Amini et al., 2011; Gribonval et al., 2012; Barsbey et al., 2021; Diao et al., 2023; Wan et al., 2024). We refer the reader to our suppl. material for a discussion / comparison.

**Dominance vs. spread**. While ($q, k, \epsilon$)-compressibility quantifies how well a vector can be approximated using its top-$k$ entries, it does not fully capture the internal structure among those dominant terms. Consider the vectors $\boldsymbol{x}_{1} = (10, 2, 1, 1)$ and $\boldsymbol{x}_{2} = (6, 6, 1, 1)$: both yield the same 2-term relative approximation error under $q = 1$, yet their dominant components differ markedly in structure. To formalize this distinction, we introduce the **spread variable** as a complementary descriptor. Given a vector $\boldsymbol{\theta}$ with elements sorted by magnitude, we define its *spread* $\beta \in [0, 1]$ via the relation $|\theta_{k}| = (1 - \beta)|\theta_{1}|$. Intuitively, $\beta$ quantifies the relative decay from the largest to the $k$-th largest entry, capturing an additional degree of freedom in the geometry of compressibility, better describing and distinguishing compressible distributions beyond what is possible with approximation error alone.

**Structured compressibility**. Importantly, given that the $\boldsymbol{\theta}$ can be any vector, the above definition can be used flexibly to describe different notions of compressibility, including those of structured compressibility, where particular substructures in the model dominate the rest. More specifically, given a layer parameter matrix $\mathbf{W} \in \mathbb{R}^{h \times h}$ from Eq (2), let $\boldsymbol{\nu} := (\|\mathbf{w}_{1}\|_{1}, \dots, \|\mathbf{w}_{h}\|_{1})$ denote $\ell_{1}$ norms of rows of the matrix $\mathbf{W}$. The compressibility of $\boldsymbol{\nu}$ would correspond to *row/neuron compressibility*, which is a desirable property for neural network parameters as it expedites pruning of whole neurons, with tangible computational gains. Note that this also would correspond to filter compressibility/pruning in CNNs with a matricization of the convolution tensor. Similarly, let $\boldsymbol{\sigma} := (\sigma_{1}, \sigma_{2}, \dots)$ denote the singular values of matrix $\mathbf{W}$. Compressibility of $\boldsymbol{\sigma}$ would correspond to *spectral compressibility*, closely related to the notion of approximate/numerical low-rankness.

## 3 Norm-based adversarial robustness bounds

**Motivating hypothesis**. Our analysis relies on a simple intuition: Although structured (neuron, spectral) compressibility is desirable from a computational perspective, it also focuses the total energy of the parameter on a few dominant terms (rows/filters, singular values). This in turn creates a few, potent directions in the latent space and increases the operator norms of the parameters ($\ell_\infty$, $\ell_2$ operator norms respectively). This increases their sensitivity to worst-case perturbations: adversarial attacks exploiting these directions are amplified in the representation space, and can more easily disrupt the predictions of the model.

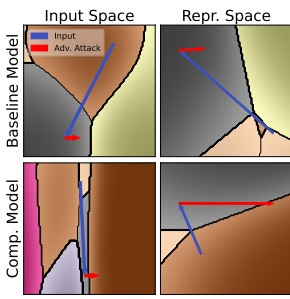

Taken from an experiment presented in full detail in Sec. 4, Fig. 1 visualizes the decision boundaries for a single sample under a baseline vs. spectrally compressible model. The figure summarizes our core hypothesis concisely: the right column shows that the representation of the adversarial perturbation in relation to that of the original image is dramatically increased in the compressible model. The attack thus successfully disrupts the prediction under compressibility, whereas it fails to do so in the baseline model. In contrast, since the attack takes place under a *fixed* budget in the input space, the reflection of this in the input space is remarkably different: decision boundaries are *contracted* around the input to reflect the vulnerability produced by the sensitive attack directions created by compressibility. Before deriving further insights from this and other experiments, we formalize our intuition in the following analysis. For brevity all proofs are deferred to the supplementary material.

Figure 1: Decision boundaries under compressibility.

**Compressibility-based Lipschitz bounds.** Our theory will relate structured compressibility to robustness through its effect on network's operator norms and Lipschitz constants. However, this brings about a particular conceptual challenge. Our notion of $(q, k, \epsilon)$ compressibility, like others' (Diao et al., 2023), is a *scale-independent* measure. Therefore, any direct relation between compressibility and Lipschitz constants would be rendered void by the arbitrary scaling of the parameters. Therefore, we characterize $\ell_\infty$ and $\ell_2$ operator norms of the parameters by an upper bound that decomposes into (compressibility $\times$ Frobenius norm) terms. This "structure vs. scale" decomposition allows us to meaningfully relate compressibility and robustness, and also allows us to develop concrete hypotheses regarding the effect of various interventions in neural network training.

**Theorem 3.1.** *The following statements relate operator norms and structured compresibility.*

*(a) Neuron compressibility (i.e. row-sparsity):* *Let $\mathbf{w}_i, i \in [h]$ denote the rows of the matrix $\mathbf{W}$, and let $\boldsymbol{\nu} := (\|\mathbf{w}_1\|_1, \ldots, \|\mathbf{w}_h\|_1)$ denote $\ell_1$ norms of its rows. Assuming $\boldsymbol{\nu}$ is $(1, k_{\boldsymbol{\nu}}, \epsilon_{\boldsymbol{\nu}})$ compressible and each row $\mathbf{w}_i$ is $(2, k_r, \epsilon_r)$ compressible implies:*

$$\|\mathbf{W}\|_\infty \leq \frac{(1 - \epsilon_{\boldsymbol{\nu}})}{(1 - \beta_{\boldsymbol{\nu}})} \left( \frac{\sqrt{h k_r} + h\epsilon_r}{k_{\boldsymbol{\nu}}} \right) \|\mathbf{W}\|_F. \tag{4}$$

*(b) Spectral compressibility (i.e. low-rankness):* *Let $\boldsymbol{\sigma} := (\sigma_1, \sigma_2, \ldots)$ denote the singular values of matrix $\mathbf{W}$. Assuming $\boldsymbol{\sigma}$ is $(1, k_{\boldsymbol{\sigma}}, \epsilon_{\boldsymbol{\sigma}})$ compressible implies:*

$$\|\mathbf{W}\|_2 \leq \frac{(1 - \epsilon_{\boldsymbol{\sigma}})}{(1 - \beta_{\boldsymbol{\sigma}})} \left( \frac{\sqrt{h}}{k_{\boldsymbol{\sigma}}} \right) \|\mathbf{W}\|_F. \tag{5}$$

Intuitively, Thm 3.1 describes how increasing compressibility affects layer operator norms: Neuron compressibility, *i.e.* a small number of rows dominating the matrix increases $\ell_\infty$ operator norm of the matrix, especially if the spread within these dominant rows are high. Similarly, increased spectral compressibility and spread increases the $\ell_2$ operator norm. Note that the latter result is closely related to results from the literature that connect stable rank or condition number to robustness (Savostianova et al., 2023; Feng et al., 2025). We highlight that although Thm 3.1 directly relates neuron and spectral compressibility to perturbations defined in $\ell_\infty$ and $\ell_2$ norms, we do not claim that relationships across attack and operator norms do not hold. Indeed in our suppl. material, we show that the two operator norms are likely to move together under compressibility, connecting structured compressibility to a broader notion of adversarial vulnerability.

As we move on to characterizing layers within a neural network, $\mathbf{W}_k^l$ will be used to denote the *compressed* version of the parameter matrix of layer $l$. In the case of row compression, this will

correspond to keeping the $k$ dominant rows as is, and setting the $h - k$ trailing rows to $\mathbf{0}$. In the case of spectral compression, given the singular value decomposition (SVD), $\mathbf{W}^l = \mathbf{U}^l \mathbf{\Sigma}^l \mathbf{V}^{l^T}$, the compressed matrix would correspond to $\mathbf{W}^l_k := \mathbf{U}^l_k \mathbf{\Sigma}^l_k \mathbf{V}^{l^T}_k$, where the $h - k$ smallest singular values are truncated.

Note that the sensitivity of the network not only relies on the characteristics of layer parameters, but also on the interactions between them. As an informative extreme case, assume that layer $\mathbf{W}^l$ greatly amplifies the input in the direction $\mathbf{u}_1$, due to spectral compressibility producing a large $\sigma_1$. Ignoring nonlinearities for now, if $\mathbf{u}_1$ is in the null space of $\mathbf{W}^{l+1}$, this amplification will have no effect on the sensitivity of the overall network. Thus, potent attack directions in the network are determined not only through layers' inherent properties, but how well the dominant directions in consecutive layers "align", in consideration with the nonlinearities between them. We will characterize this crucial interaction with the *interlayer alignment terms* $A^*_\infty$ and $A^*_2$. With $\mathcal{D}$ as the set of all diagonal binary matrices, standing for all possible ReLU activation patterns, these are defined as:

$$A^*_\infty(\mathbf{W}^{l+1}_k, \mathbf{W}^l_k) \triangleq \max_{\mathbf{D} \in \mathcal{D}} \frac{\|\mathbf{W}^{l+1}_k \mathbf{D} \mathbf{W}^l_k\|_\infty}{\|\mathbf{W}^{l+1}\|_\infty \|\mathbf{W}^l\|_\infty} + R_\infty(\epsilon) \tag{6}$$

$$A^*_2(\mathbf{W}^{l+1}_k, \mathbf{W}^l_k) \triangleq \max_{\mathbf{D} \in \mathcal{D}} \frac{\|\sqrt{\Sigma^{l+1}_k} \mathbf{V}^{l+1^T}_k \mathbf{D} \mathbf{U}^l_k \sqrt{\Sigma^l_k}\|_2}{\sqrt{\|\mathbf{W}^{l+1}\|_2 \|\mathbf{W}^l\|_2}} + R_2(\epsilon), \tag{7}$$

where $R_\infty(\epsilon) := \nu^l_{k+1}/\nu^l_1 + \nu^{l+1}_{k+1}/\nu^{l+1}_1 + \nu^l_{k+1}\nu^{l+1}_{k+1}/\nu^l_1\nu^{l+1}_1$ is a remainder alignment term and likewise, $R_2(\epsilon) := \sqrt{\sigma^l_k/\sigma^l_1} + \sqrt{\sigma^{l+1}_{k+1}/\sigma^{l+1}_1} + \sqrt{\sigma^l_{k+1}\sigma^{l+1}_{k+1}/\sigma^l_1\sigma^{l+1}_1}$. In the suppl. material, we show that for $p \in \{2, \infty\}$, $R_p(\epsilon) \to 0$ as $\epsilon \to 0$. There, we also show that for all layers $A^*_p \le 1$; alignment terms can therefore be interpreted to act as a normalized function that corrects the worst-case bound based on the dominant terms' misalignment. Next theorem will use Thm 3.1 and Eq (6) and (7) to provide an upper bound to the Lipschitz constant of the network.

**Theorem 3.2.** *Let $L^p_\Phi$ be the Lipschitz constant of the encoder $\Phi$ defined following Eq (2). Let $\mathcal{D}$ denote the set of all diagonal binary matrices, corresponding to ReLU activation layers. Then:*

*(a) Row/neuron compressibility: The $\ell_\infty$ Lipschitz constant of $\Phi$ can be upper bounded by:*

$$L^\infty_\Phi \le \hat{L}^\infty_\Phi := \prod_{l=1}^\lambda \frac{(1 - \epsilon_{\boldsymbol{\nu}})}{(1 - \beta_{\boldsymbol{\nu}})} \left( \frac{\sqrt{hk_r} + h\epsilon_r}{k_{\boldsymbol{\nu}}} \right) \|\mathbf{W}\|_F \prod_{l=1}^{\lambda-1} \tilde{A}^*_\infty(\mathbf{W}^{\{l+1\}}_k, \mathbf{W}^l_k), \tag{8}$$

*where $\tilde{A}^*_\infty(\mathbf{W}^{\{l+1\}}_k, \mathbf{W}^l_k) = A^*_\infty(\mathbf{W}^{\{l+1\}}_k, \mathbf{W}^l_k)$ if $l \in S_{opt}$, and 1 otherwise. $S_{opt} \subseteq \{1, 2, \ldots, L-1\}$ is the optimal alignment partition set (See Dfn. A.4) that can be determined in $O(\lambda)$ time.*

*(b) Spectral compressibility: The $\ell_\infty$ Lipschitz constant of $\Phi$ can be upper bounded by:*

$$L^2_\Phi \le \hat{L}^2_\Phi := \prod_{l=1}^\lambda \frac{(1 - \epsilon_{\boldsymbol{\sigma}})}{(1 - \beta_{\boldsymbol{\sigma}})} \left( \frac{\sqrt{h}}{k_{\boldsymbol{\sigma}}} \right) \|\mathbf{W}\|_F \prod_{l=1}^{\lambda-1} A^*_2(\mathbf{W}^{\{l+1\}}_k, \mathbf{W}^l_k). \tag{9}$$

We note that this upper bound can be directly used in conjunction with other results from the literature (Ribeiro et al., 2023) to characterize adversarial robustness gap:

**Corollary 3.3.** *Under a binary classification task with cross-entropy loss, $\ell(y, \boldsymbol{x}^\top \boldsymbol{\theta}) = \ell(y, \hat{y}) = \log\left(1 + e^{-y\hat{y}}\right)$, given a neural network classifier as described in (2), under the same assumptions with (8), $F^{adv}_\infty(\boldsymbol{\theta}; \delta) \le F(\boldsymbol{\theta}) + \delta \hat{L}^\infty_\Phi \|\boldsymbol{\theta}\|_1$. Similarly, under the assumptions of $F^{adv}_2(\boldsymbol{\theta}; \delta) \le F(\boldsymbol{\theta}) + \delta \hat{L}^2_\Phi \|\boldsymbol{\theta}\|_2$.*

Note that although bounds provided in Thm 3.2 are tighter than the pessimistic "product-of-norms" bounds, it deliberately *trades off* some tightness by utilizing Thm 3.1. However, in return, this results in a bound that decomposes into analytically interpretable and actionable terms. Such bounds have proven valuable in analyzing adversarial robustness in deep learning (Wen et al., 2020). Regardless, in our suppl. material we show that our bounds correlate with adversarial robustness gap, as well as showing that as the global Lipschitz constant increases, empirically estimated local Lipschitz constants scale accordingly. There, we also explore the alignment terms' empirical behavior and estimation techniques, although a detailed analysis thereof lies beyond our primary focus. We now translate these theoretical insights into concrete hypotheses and test them through experiments.

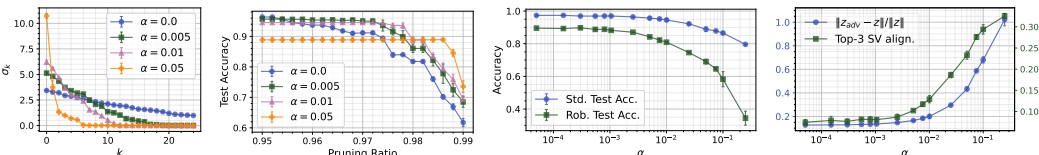

Figure 2: Model statistics under increasing strength of nuclear norm regularization ($\alpha$).

## 4 Experimental evaluation

We now validate our theoretical findings through specific experimentation. We first validate our *motivating hypothesis* and then empirically show that (i) neuron and spectral compressibility-inducing interventions will reduce adversarial robustness against $\ell_\infty$ and $\ell_2$ adversarial attacks; (ii) the negative effects of compressibility to persist under adversarial training, (iii) the compressibility-related vulnerabilities being baked into the learned representations during pretraining, will impact any downstream task in transfer learning; (iv) increasing compressibility creates vulnerable directions in the latent space, further enabling universal adversarial examples (UAEs), while increasing Frobenius norm will create vulnerability without leading to UAEs; and (v) compressed models will inherit the vulnerability of the original models, and conducting compression based on $(q, k, \epsilon)$ -compressibility and reducing the spread of the dominant terms will improve robustness.

**Datasets, architectures, and training**. We conduct our experiments in the most commonly used datasets and architectures in the literature on adversarial robustness under pruning (Piras et al., 2024). Datasets we use include MNIST (Deng, 2012), CIFAR-10, CIFAR-100 (Krizhevsky & Hinton, 2009), SVHN (Netzer et al., 2011). Architectures we utilize include fully connected networks (FCN), ResNet18 (He et al., 2016), VGG16 (Simonyan & Zisserman, 2014), and WideResNet-101-2 (Zagoruyko & Komodakis, 2016). Unless otherwise noted, we use softmax cross-entropy loss, the AdamW optimizer with a weight decay of $0.01$, a learning rate of $0.001$, and use validation set based model selection for early stopping.

**Evaluating and training for adversarial robustness**. When evaluating adversarial robustness, we utilize AutoPGD as the primary adversarial attack algorithm for evaluation (Croce & Hein, 2020), through its implementation by Nicolae et al. (2018). When training for adversarial robustness, we utilize a PGD attack to generate adversarial samples at every iteration (Madry et al., 2018). Unless otherwise noted, we use a ratio of 0.5 for adversarial samples in a training minibatch. We use $\epsilon = 8/255$ and $\epsilon = 0.5$ for $\ell_\infty$ and $\ell_2$ attacks respectively for end-to-end adversarially trained models. We use $0.25$ of these budgets for standard trained or adversarially fine-tuned models to allow a visible comparison. By default, we present results for $\ell_\infty$ and $\ell_2$ attacks when evaluating robustness under neuron and spectral compressibility respectively, and defer the cross-norm results to the supplementary material, which also includes further details on our experiment settings and implementation.

### 4.1 Results

**Testing the motivating hypothesis** We start our empirical analysis with a demonstrative experiment to visually investigate the implications of our initial hypothesis. For this, we train a single 400-width hidden layer FCN with ReLU activations on the MNIST dataset. We use nuclear norm regularization (NNR) to encourage singular value (SV) compressibility, adding the term $\alpha\|\boldsymbol{\sigma}\|_1$ to the training objective, with $\alpha$ as a hyperparameter. To avoid confounding by NNR decreasing overall parameter norms, we apply Frobenius norm normalization to $\mathbf{W}^1$ at every iteration (Miyato et al., 2018).

In Fig. 2 (left) we validate that our intervention indeed increases spectral norm compressibility. As expected, Fig. 2 (center left) shows that SV compressibility actually allows pruning: the more compressible models retain their performance under stronger pruning. Fig. 2 (center right) shows that increased compressibility comes at the cost of adversarial robustness: as $\alpha$ increases, adversarial accuracy dramatically falls. We further investigate whether this fall is due to our hypothesized mechanism. Let $\boldsymbol{z} = \Phi(\boldsymbol{x})$ and $\boldsymbol{z}_{\text{adv}} = \Phi(\boldsymbol{x} + \boldsymbol{a}^*)$ denote the learned representations of clean and perturbed input images. If the adversarial attacks are taking advantage of the potent directions created by compressibility, then as compressibility increases: (1) The perturbations $\boldsymbol{a}^*$ should align more with the dominant singular directions, *i.e.*, $\mathbf{v}_i^\mathsf{T}\boldsymbol{a}^* \gg \mathbf{v}_j^\mathsf{T}\boldsymbol{a}^* \; \forall i \in [k], j \notin [k]$, (2) representations of adversarial perturbations should grow stronger in relation to the original image's representation, *i.e.*

273 $\|z_{\text{adv}} - z\|_2/\|z\|_2$ should increase. Fig. 2 (right) confirms both predictions. Lastly, the previously
274 presented Fig. 1 visualizes the effect of compressibility in the input and representation space.

275 **Adversarial robustness and com-**
276 **pressibility under standard training.**
277 For implications of our analysis un-
278 der more realistic settings, we start
279 by investigating the effects of com-
280 pressibility on adversarial robustness
281 in fully connected networks (FCN).
282 We induce neuron and spectral com-
283 pressibility through group lasso regu-
284 larization[1] and low-rank factorization,
285 respectively (latter avoids the exces-
286 sive cost of nuclear norm regulariza-
287 tion). As above, we conduct Frobe-
288 nius norm normalization at every iter-
289 ation. Fig. 3 (top) presents the results
290 of these experiments: The reduction
291 in adversarial robustness as a function
292 of increasing compressibility is clear
293 in both cases, confirming our main hy-

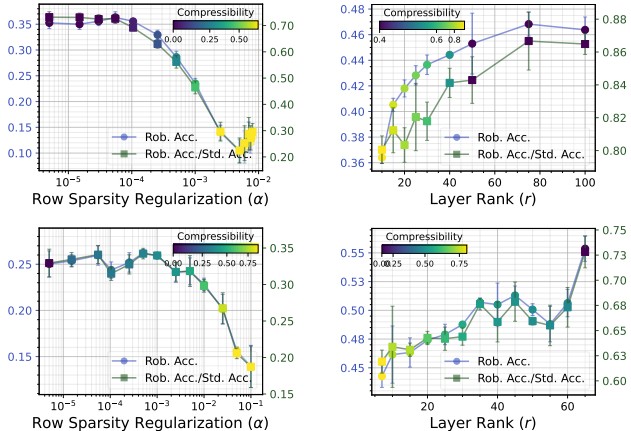

Figure 3: Results with FCN (top) and ResNet18 (bottom) trained on CIFAR-10 dataset.

294 pothesis. Note that we present robust accuracy / standard accuracy ratio alongside robust accuracy
295 to highlight that the obtained results are not due to baseline standard accuracy being lower under
296 compressibility.

297 We then investigate whether our hypotheses apply beyond the context of our theory, and test our
298 predictions in ResNet18 models trained on CIFAR-10 datasets. Here we eschew Frobenius norm
299 normalization for standard weight decay. However, to prevent confounding from group lasso's effect
300 on general parameter scales, we create a scale-invariant version that regularizes row norms' $\ell_1/\ell_2$
301 norm ratio.[2] Fig. 3 (bottom) demonstrates that the effects describe above clearly translate to this
302 setting as well, further solidifying the relationship between structured compressibility and adversarial
303 robustness. We present similar results on two other architectures (VGG16, WideResNet-101) and
304 two other datasets (CIFAR-100, SVHN) in the suppl. material. Going forward, for brevity we will
305 focus on neuron compressibility results, and defer corresponding spectral compressibility results to
306 the suppl. material, where we also discuss unstructured compressibility and inductive-bias based
307 emergent compressibility.

308 **Effect of adversarial training / fine-**
309 **tuning on network compressibility.**
310 Given that adversarial training is the
311 primary method for obtaining models
312 that are robust against adversaries, we
313 next investigate whether the effects
314 we have observed will persist under
315 this regime. We first take two models
316 from the setting presented in Fig. 3
317 with ResNet18s trained on CIFAR-10

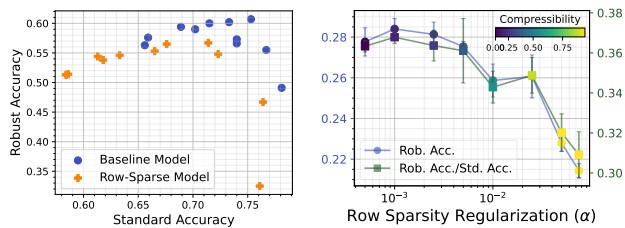

Figure 4: Adversarial fine-tuning and training.

318 under row sparsity regularization, and take the baseline model as well as a model with high sparsity
319 regularization (regularization parameter = 0.05). Afterwards, we fine-tune both models for 10 epochs
320 under adversarial training, using various adversarial sample ratios $\in [0, 1]$. The results are presented
321 in Fig. 4 (left), and show a remarkable pattern: While both models demonstrate a large variability
322 in terms of robust vs. standard accuracy trade-off based on the sample ratio in the fine-tuning, the
323 original difference in their robustness performance remains, as the different versions of the two
324 models form two pareto fronts. Next, we investigate whether the impact of compressibility on robust-
325 ness will disappear under adversarial training from initialization. To make this setting as close to
326 practice as possible, we also include a learning rate annealing schedule (Cosine annealing) and basic

---

[1]Group lasso regularization penalizes the $\ell_1$ norm of row $\ell_2$ norms of each layer, promoting row-sparsity.

[2]In the suppl. material, we show that standard group lasso creates a "tug-of-war" between increasing compressibility and decreasing parameter scales; the former eventually wins, resulting in decreased robustness.

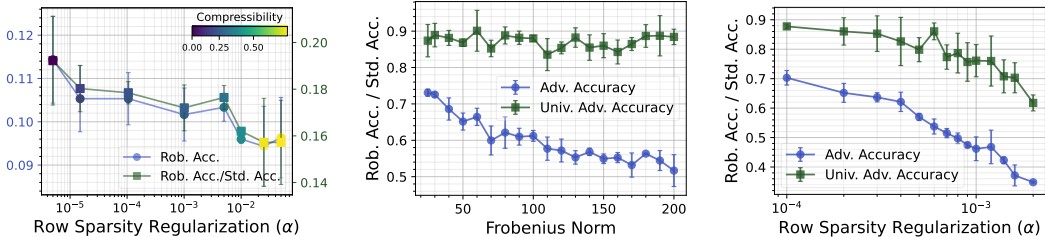

Figure 5: Transfer learning and universal adversarial examples.

data augmentation (horizontal flip and random crops). The results almost identically replicate our observations under standard training. Although adversarial training increases adversarial robustness overall, the relative effect of compressibility remains as is.

**Transferability of adversarial vulnerability**. Next, we investigate our hypothesis that the effects of compressibility should persist under transfer learning due to the structural effects created on representations. We train a ResNet18 model on CIFAR-100 dataset with increasing row sparsity regularization. After the training is complete, we train a linear classifier head for prediction on CIFAR-10 dataset and evaluate the robustness of the resulting model. Fig. 5 (left) shows that the effects of compressibility observed above directly translate to the context of transfer learning, where increased compressibility in pretraining affects robustness performance in the downstream task, for which the network is fine-tuned.

**Universal adversarial examples**. Examining the terms in Thm 3.2, we predict that while both compressibility and Frobenius norm are likely to increase vulnerability, only the former is likely to lead universal adversarial examples (UAEs) (Moosavi-Dezfooli et al., 2017), due to the global vulnerable directions it creates. To test our hypothesis, we modify the setting of FCN experiments presented above: As an alternative to increasing row sparsity regularization under a fixed Frobenius norm, in an alternative set of experiments we systematically increase the constant to which Frobenius norm of the layers is fixed, without any row sparsity regularization. We utilize a FGSM-based (Goodfellow et al., 2015) UAE computation to develop adversarial samples. Fig. 5 (center, right) confirms our hypothesis: while increasing Frobenius norm only decreases standard adversarial robustness, increasing compressibility additionally creates vulnerability to UAEs.

**Compression and robustness**. We now investigate whether the compressed models inherit the adversarial vulnerability of the original models, as they are subjected to increasing layerwise filter pruning. Using the ResNet18 and CIFAR-10 combination under adversarial training, in Fig. 6 (left), we compare the baseline model ($\alpha = 0.0$) to a compressible model ($\alpha = 0.1$). We see that at no point the compressed model surpasses the baseline model's uncompressed performance in terms of standard and robust accuracy. However, as expected, as pruning increases the baseline model fails to retain its standard nor robust performance whereas the compressible model does considerably better, demonstrating the fundamental tension between robustness and compressibility.

Lastly, we develop two simple interventions based on our bound that results in tangible improvements in reconciling compressibility and robustness. Given the fact that layerwise pruning is known to produce harmful bottlenecks that lead to layer collapse (Blalock et al., 2020), we develop an intuitive global compression method based on our bound. Instead of targeting a pruning ratio and pruning each layer accordingly, we set a target $\epsilon$, and for each layer compute $k$ that satisfies this $\epsilon$ level. Given a target global pruning ratio, we scan over different levels of $\epsilon$ and determine the level that gets closest to the target ratio. Moreover, during training we control the spread of the dominant terms, $\beta$, which our analyses show to be harmful for robustness, without increasing compressibility. We accomplish this through simply regularizing the variance of the top $0.05$ of each layer's

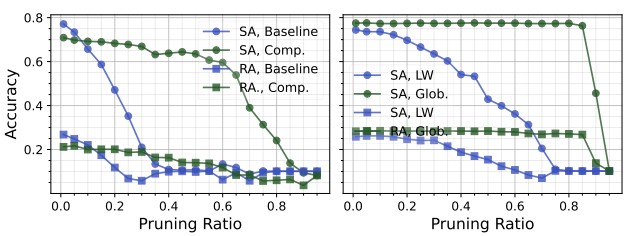

Figure 6: Robustness under compression. SA/RA: Standard/Robust Acc. LW/Glob.: Layerwise vs. global pruning.

filters' norms. Fig. 6 (right) demonstrates that our interventions create a dramatic improvement in performance retention, demonstrated on a model with $\alpha = 0.01$.

# 5 Related work

**Adversarial robustness**. The susceptibility of the neural network models to adversarial examples created through small perturbations (Szegedy et al., 2014) engendered a lot of research investigating the issue (Madry et al., 2018). To this day adversarial robustness remains one of the most important topics in machine learning security (Malik et al., 2024). The literature ranges from the development of new attacks and defenses (Moosavi-Dezfooli et al., 2016; Abdollahpoorrostam et al., 2024), to investigating sources/mechanisms of adversarial vulnerability, to implications of AEs for the inductive biases of modern machine learning architectures (Ilyas et al., 2019; Ortiz-Jimenez et al., 2021; Xu et al., 2024), to developing strategies to retain model expressivity and generalization while defending against adversarial attacks (Tsipras et al., 2019; Zhang et al., 2024).

**Compressibility and pruning**. Prominent compression approaches include pruning, quantization, distillation, conditional computing, and efficient architecture development (O'Neill, 2020). Out of these, pruning remains among the most actively researched compression approaches due to its versatility (Cheng et al., 2024). Inducing compressibility / sparsity at training time is the easiest way to obtain prunable models (Hohman et al., 2024). Compressibility across different substructures, a.k.a group sparsity (Li et al., 2020b), allows for structured pruning (e.g. neuron/row, filter/channel, kernel pruning), which is computationally efficient (Yang et al., 2018), yet lead to sharp reduction in network connectivity, threatening performance (Blalock et al., 2020). Lastly, spectral compressibility relaxes the notion of low-rankness, utilized for approximating large matrices with appealing theoretical properties (Suzuki et al., 2020; Schotthöfer et al., 2022).

**Compressibility and robustness**. Whereas some research argues that compressibility is beneficial for adversarial robustness (Guo et al., 2018; Balda et al., 2020; Liao et al., 2022), others indicate the relation is *at best* highly dependent on the degree and type of compressibility, as well as attack type (Li et al., 2020a; Merkle et al., 2022; Savostianova et al., 2023; Feng et al., 2025). While a stream of new methods that incorporate adversarial robustness in novel ways to pruning, newly emerging systematic benchmarks reveal at best marginal benefits for such methods compared to weight-based pruning (Lee et al., 2020; Piras et al., 2024). Whereas some methods demonstrate benefits of adversarial training-aware sparsification (Gui et al., 2019; Sehwag et al., 2020; Pavlitska et al., 2023), infamous problems adversarial training (AT) poses for standard generalization, transferability, as well as computational feasibility especially for larger models still plague such methods (Tsipras et al., 2019; Wen et al., 2020; Yang et al., 2024).

# 6 Conclusion and future work

In this paper, we present a unified theoretical and empirical treatment of how structured compressibility shapes adversarial robustness. Via a novel analysis of neuron-level and spectral compressibility, we uncover a fundamental mechanism: compression concentrates sensitivity along a small number of directions in representation space, rendering models more vulnerable—even under adversarial training and transfer learning. Our norm-based robustness bounds offer interpretable decompositions that predict both standard and universal adversarial vulnerability, and shed light on the trade-offs between efficiency and security in modern neural networks. Empirically, we validate these insights across datasets, architectures, and training regimes, showing how both compressibility and its spread determines adversarial susceptibility. We show that these vulnerabilities can be mitigated through targeted strategies guided by our bounds.

Our work provides a novel insight into the relationship between structured compressibility and adversarial vulnerability. A limitation is our theory's reliance on global Lipschitz constants to characterize network performance: future work should focus on providing a unified view that incorporates both structural/global weaknesses, as well as the localization of sensitivity in the input space. Moreover, while the simple interventions suggested by our theory provides cost-effective improvements to the compressibility-robustness trade-off, these insights should be combined with novel compression methods to improve the frontiers of robust compression.

**Broader impact statement**. Our research is largely theoretical and raises no direct societal or ethical concerns. To the extent that it has any downstream effects, we expect them to be positive by increasing robustness and resource-efficiency of machine learning models.

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

# On the Interaction of
# Compressibility and Adversarial Robustness
# –Supplementary Material–

## Contents

## A Proofs

We start with a number of auxiliary results that are used in the theorems and corollary presented in Sec. 3.

**Lemma A.1.** *For any strictly $(q, k, \epsilon)$ compressible vector $\boldsymbol{\theta}$ and for all $q \geq 1$, $\|\boldsymbol{\theta}^{(k)}\|_q = (1 - \epsilon^q)^{1/q} \|\boldsymbol{\theta}\|_q$.*

*Proof.* $\|\boldsymbol{\theta} - \boldsymbol{\theta}^{(k)}\|_q^q = \epsilon^q \|\boldsymbol{\theta}\|_q^q$ follows from the definition of compressibility. Adding $\|\boldsymbol{\theta}^{(k)}\|_q^q$ to both sides leads to $\|\boldsymbol{\theta}\|_q^q = \epsilon^q \|\boldsymbol{\theta}\|_q^q + \|\boldsymbol{\theta}^{(k)}\|_q^q$, with LHS due to elements of $\boldsymbol{x}$ and $\boldsymbol{\theta} - \boldsymbol{\theta}^{(k)}$ populating disjoint sets of coordinates. Result follows with simple algebraic manipulation. $\square$

Note that for the results in this section, we use $\boldsymbol{\theta}^{(k)}$ and $\boldsymbol{\theta}_k$ equivalently to denote a vector that includes only the $k$ dominant terms.

**Lemma A.2.** *For $p^* < q$, given the $(2, k, \epsilon)$-compressible vector $\boldsymbol{\theta} \in \mathbb{R}^d$, we have:*

$$\|\boldsymbol{\theta}\|_{p^*} \leq k^{\frac{1}{p^*} - \frac{1}{q}} \|\boldsymbol{\theta}^{(k)}\|_q + d^{\frac{1}{p^*} - \frac{1}{q}} \epsilon \|\boldsymbol{\theta}\|_q. \tag{10}$$

*Proof.* We start by applying Minkowsky's inequality to $\|\boldsymbol{\theta}\|_{p^*}$:

$$\|\boldsymbol{\theta}\|_{p^*} \leq \|\boldsymbol{\theta}^{(k)}\|_{p^*} + \|\boldsymbol{\theta} - \boldsymbol{\theta}^{(k)}\|_{p^*}. \tag{11}$$

We now bound the terms on RHS separately. For the first term, since $p^* < q$ by Hölder's inequality for k-sparse vectors we have

$$\|\boldsymbol{\theta}^{(k)}\|_{p^*} \leq k^{\frac{1}{p^*} - \frac{1}{q}} \|\boldsymbol{\theta}^{(k)}\|_q.$$

For the next term, we can write

$$\|\boldsymbol{\theta} - \boldsymbol{\theta}^{(k)}\|_{p^*} \leq d^{\frac{1}{p^*} - \frac{1}{q}} \|\boldsymbol{\theta} - \boldsymbol{\theta}^{(k)}\|_q \leq d^{\frac{1}{p^*} - \frac{1}{q}} \epsilon \|\boldsymbol{\theta}\|_q,$$

with the left inequality due to Hölder's inequality, and the right due to $\boldsymbol{\theta}^{(k)}$'s $(q, k, \epsilon)$ compressibility. Combining the expressions for both terms, we have

$$\|\boldsymbol{\theta}\|_{p^*} \leq k^{\frac{1}{p^*} - \frac{1}{q}} \|\boldsymbol{\theta}^{(k)}\|_q + d^{\frac{1}{p^*} - \frac{1}{q}} \epsilon \|\boldsymbol{\theta}\|_q. \tag{12}$$

$\square$

**Proposition A.3.** *Given a linear binary classifier and binary cross-entropy loss function, we have the following bound:*

$$F_p^{\mathrm{adv}}(\boldsymbol{\theta}; \delta) \leq F(\boldsymbol{\theta}; \delta) + \delta \|\boldsymbol{\theta}\|_{p^*} \tag{13}$$

*Proof of Proposition A.3.* For binary cross-entropy loss we have:

$$f^{\mathrm{adv}}(\boldsymbol{x}, \boldsymbol{\theta}; \delta) = \log\left(1 + \exp\left(-y(\boldsymbol{x}^\top \boldsymbol{\theta}) + \delta \|\boldsymbol{\theta}\|_{p^*}\right)\right).$$

We observe that $f^{\mathrm{adv}}(\boldsymbol{x}, \boldsymbol{\theta}; \delta) \leq f(\boldsymbol{x}, \boldsymbol{\theta}; \delta) + \delta \|\boldsymbol{\theta}\|_{p^*}$ since

$$
\begin{aligned}
f^{\mathrm{adv}}(\boldsymbol{x}, \boldsymbol{\theta}; \delta) &= \log\left(1 + \exp\left(-y(\boldsymbol{x}^\top \boldsymbol{\theta}) + \delta \|\boldsymbol{\theta}\|_{p^*}\right)\right) \\
&= \log\left(1 + \exp\left(-y(\boldsymbol{x}^\top \boldsymbol{\theta})\right)\right) + \log\left(\frac{1 + \exp\left(-y(\boldsymbol{x}^\top \boldsymbol{\theta}) + \delta \|\boldsymbol{\theta}\|_{p^*}\right)}{1 + \exp\left(-y(\boldsymbol{x}^\top \boldsymbol{\theta})\right)}\right) \\
&= f(\boldsymbol{x}, \boldsymbol{\theta}; \delta) + \log\left(1 + (\exp\left(\delta \|\boldsymbol{\theta}\|_{p^*}\right) - 1)\frac{\exp\left(-y(\boldsymbol{x}^\top \boldsymbol{\theta})\right)}{1 + \exp\left(-y(\boldsymbol{x}^\top \boldsymbol{\theta})\right)}\right) \\
&\leq f(\boldsymbol{x}, \boldsymbol{\theta}; \delta) + \delta \|\boldsymbol{\theta}\|_{p^*},
\end{aligned}
$$

with the last inequality due to the fact that $\frac{\exp\left(-y(\boldsymbol{x}^\top \boldsymbol{\theta})\right)}{1 + \exp\left(-y(\boldsymbol{x}^\top \boldsymbol{\theta})\right)} < 1$. Taking the expectation of the expression gives:

$$F^{\mathrm{adv}}(\boldsymbol{\theta}; \delta) \leq F(\boldsymbol{\theta}; \delta) + \delta \|\boldsymbol{\theta}\|_{p^*}$$

$\square$

**Main results**. We now present the proofs for Thm. 3.1 and 3.2 and Corollary 3.3.

*Proof of Thm 3.1.* For brevity we will omit $\boldsymbol{\nu}$ as a subscript, such that $\epsilon = \epsilon_{\boldsymbol{\nu}}, k = k_{\boldsymbol{\nu}}, \beta = \beta_{\boldsymbol{\nu}}$.

For **(a)**, we assume $\boldsymbol{\nu}$ is in a descending order w.l.o.g., and $\hat{\boldsymbol{\nu}}$ is the corresponding vector of $\ell_2$ norms for each row. We note that

$$\|\boldsymbol{\nu}^{(k)}\|_1 = \sum_{i=1}^{k} \nu_i \geq k\nu_k \tag{14}$$

$$\geq k(1-\beta)\nu_1 \tag{15}$$

$$(1-\epsilon)\|\boldsymbol{\nu}\|_1 \geq k(1-\beta)\nu_1 \tag{16}$$

$$\frac{(1-\epsilon)}{(1-\beta)}\frac{1}{k}\|\boldsymbol{\nu}\|_1 \geq \nu_1 \tag{17}$$

$$\frac{(1-\epsilon)}{(1-\beta)}\frac{1}{k}\|\boldsymbol{\nu}\|_1 \geq \|\mathbf{W}\|_\infty \tag{18}$$

with (14) being the smallest magnitude element in $\boldsymbol{\nu}^{(k)}$, (15) due to the definition of slack variable $\beta$, and (16) due to Lemma A.1, and (18) due to the fact that $\|\mathbf{W}\|_\infty = \nu_1$, as $\boldsymbol{\nu}$ is assumed to be magnitude-ordered. We then move on to characterizing $\|\boldsymbol{\nu}\|_1$. Notice that

$$\|\boldsymbol{\nu}\|_1 = \sum_{i=1}^{h} \nu_i \leq \sum_{i=1}^{h} \sqrt{h}\hat{\nu}_i \tag{19}$$

$$\leq \sqrt{h}\|\hat{\boldsymbol{\nu}}\|_1 \tag{20}$$

$$\leq \sqrt{h}\left(\sqrt{k_r}\|\hat{\boldsymbol{\nu}}^{(k_r)}\|_2 + \sqrt{h}\|\hat{\boldsymbol{\nu}}\|_2\right) \tag{21}$$

$$\leq \left(\sqrt{hk_r} + \sqrt{h}\epsilon_r\right)\|\hat{\boldsymbol{\nu}}\|_2 \tag{22}$$

$$\leq \left(\sqrt{hk_r} + \sqrt{h}\epsilon_r\right)\|\mathbf{W}\|_F \tag{23}$$

Note that (19) is due to standard norm inequality between $\ell_1$ and $\ell_2$ rows, (21) is due to Lemma A.2, and (23) is due to $\ell_2$ norm of the vector of row $\ell_2$ rows equals the Frobenius norm. Plugging (23) back into (18) gives the desired result.

For **(b)** the proof follows similarly through steps (14)-(17) by replacing $\boldsymbol{\nu}$ with $\boldsymbol{\sigma}$. After that, we continue with

$$\frac{(1-\epsilon)}{(1-\beta)}\frac{1}{k}\|\boldsymbol{\sigma}\|_1 \geq \sigma_1 \tag{24}$$

$$\frac{(1-\epsilon)}{(1-\beta)}\frac{1}{k}\|\boldsymbol{\sigma}\|_1 \geq \|\mathbf{W}\|_2 \tag{25}$$

$$\frac{(1-\epsilon)}{(1-\beta)}\frac{\sqrt{h}}{k}\|\boldsymbol{\sigma}\|_2 \geq \|\mathbf{W}\|_2 \tag{26}$$

$$\frac{(1-\epsilon)}{(1-\beta)}\frac{\sqrt{h}}{k}\|\mathbf{W}\|_F \geq \|\mathbf{W}\|_2 \tag{27}$$

with (25) due to $\|\mathbf{W}\|_2 = \sigma_1$, (26) due to standard norm inequality between $\ell_1$ and $\ell_2$ norms, and (27) due to the fact that $\ell_2$ norm of singular values equals Frobenius norm, i.e. $\|\mathbf{W}\|_F = \|\boldsymbol{\sigma}\|_2$. □

*Proof of Thm 3.2.* Proofs for both conditions rely on an additive decomposition of the layer matrices $\mathbf{W}^l$ into dominant/leading terms vs. remainder terms, i.e. $\mathbf{W}^l = \mathbf{W}_k^l + \mathbf{W}_r^l$. In structured compressibility this takes the form of $\mathbf{W}_k^l$ and $\mathbf{W}_r^l$ including $k$ leading (largest $\ell_1$ norm) rows and $h - k$ remaining rows, respectively, with the rest of the rows set to $\mathbf{0}$ in both cases. In spectral compressibility, this takes the form of $\mathbf{W}_k^l + \mathbf{W}_r^l = \mathbf{U}_k^l \boldsymbol{\Sigma}_k^l (\mathbf{V}_k^l)^\top + \mathbf{U}_r^l \boldsymbol{\Sigma}_r^l (\mathbf{V}_r^l)^\top$, where the remaining $h - k$ vs. leading $k$ singular values are set to 0 respectively.

Let $\boldsymbol{z}^l$ denote the post-activation representations of the network after layer $l \in [\lambda]$. The Jacobian of the network output $\boldsymbol{z}^\lambda$ with respect to the input $\boldsymbol{x}$ is given by:

$$\mathbf{J}_\Phi(\boldsymbol{x}) = \mathbf{D}^\lambda(\boldsymbol{x})\mathbf{W}^\lambda \mathbf{D}^{\lambda-1}(\boldsymbol{x})\mathbf{W}^{\lambda-1}\mathbf{D}^{\lambda-2}(\boldsymbol{x})\ldots\mathbf{D}^1(\boldsymbol{x})\mathbf{W}^1, \tag{28}$$

where $\mathbf{D}^l(\boldsymbol{x})$ is the diagonal binary matrix corresponding to the ReLU activation after layer $l$, i.e., $(\mathbf{D}^l)_{ii} = \mathbb{I}[(\bar{\boldsymbol{z}}^l)_i > 0]$, with $\bar{\boldsymbol{z}}^l$ being the pre-activation representation at layer $l$ for input $\boldsymbol{x}$.

Letting $L_\Phi^p$ denote the $p$-norm Lipschitz constant of the compressed encoder in the input domain, it can be computed as the maximum $p \to p$ operator norm of the Jacobian over the input space $\mathcal{X}$:

$$L_\Phi^p = \sup_{\boldsymbol{x}\in\mathcal{X}} \|\mathbf{J}_\Phi(\boldsymbol{x})\|_p = \sup_{\boldsymbol{x}\in\mathcal{X}} \|\mathbf{D}^\lambda(\boldsymbol{x})\mathbf{W}^\lambda \mathbf{D}^{\lambda-1}(\boldsymbol{x})\mathbf{W}^{\lambda-1}\ldots\mathbf{D}^1(\boldsymbol{x})\mathbf{W}^1\|_p. \tag{29}$$

For brevity, we use the following notation:

$$\mathbf{P}(\mathbf{D}) := \mathbf{D}^\lambda(\boldsymbol{x})\mathbf{W}^\lambda \mathbf{D}^{\lambda-1}(\boldsymbol{x})\mathbf{W}^{\lambda-1}\ldots\mathbf{D}^1(\boldsymbol{x})\mathbf{W}^1. \tag{30}$$

Note that the optimization over $\mathcal{X}$ can be replaced with the optimization over all binary activation matrices $\mathbf{D}^l \in \mathcal{D}$ for each layer whenever convenient. We replace the notation $\mathbf{D}^l(\boldsymbol{x})$ with $\mathbf{D}^l$ when doing so.

For brevity, we introduce the following abbreviations for the alignment terms with a slight abuse of notation:

$$A_{p,l}^* := A_p^*(\mathbf{W}^{l+1},\mathbf{W}^l) := \max_{\mathbf{D}\in\mathcal{D}} A_{p,l} := \max_{\mathbf{D}\in\mathcal{D}} A_p(\mathbf{W}^{l+1},\mathbf{D},\mathbf{W}^l), \tag{31}$$

where $A_p(\mathbf{W}^{l+1},\mathbf{D},\mathbf{W}^l)$ is the inner RHS term optimized over in Eq (6) and Eq (7).

**(a) Row/neuron compressibility** We aim to bound $L_\Phi^\infty$ as:

$$L_\Phi^\infty \leq \max_{\mathbf{D}^1,\ldots,\mathbf{D}^\lambda} \|\mathbf{P}(\mathbf{D})\|_\infty. \tag{32}$$

We start by noting that we can upper bound this norm by partitioning the inside terms based on the submultiplicative property:

$$\|\mathbf{P}(\mathbf{D})\|_\infty \leq \|\mathbf{D}^\lambda\mathbf{W}^\lambda\mathbf{D}^{\lambda-1}\mathbf{W}^{\lambda-1}\ldots\mathbf{D}^1\mathbf{W}^1\|_\infty \tag{33}$$

$$\leq \|\mathbf{W}^\lambda\mathbf{D}^{\lambda-1}\mathbf{W}^{\lambda-1}\|_\infty\|\mathbf{D}^{\lambda-2}\|_\infty\|\mathbf{W}^{\lambda-2}\|_\infty$$
$$\ldots\|\mathbf{W}^{l+1}\mathbf{D}^l\mathbf{W}^l\|_\infty\ldots\|\mathbf{D}^1\|_\infty\|\mathbf{W}^1\|_\infty \tag{34}$$

Note that any such parsing is valid as long as a layer does not appear in two interlayer terms at once. Given a valid parsing set $S \subseteq \{1, 2, \ldots, \lambda-1\}$, we have the interlayer alignment terms for $l \in S$, i.e. $\|\mathbf{W}^{l+1}\mathbf{D}^l\mathbf{W}^l\|_\infty$ and standalone terms for all remaining layers $\{l \mid l \notin S, l+1 \notin S\}$: $\|\mathbf{W}^l\|_\infty$. We denote all such valid parsing layer subsets with $\mathcal{S}$, where $S$ does not include any consecutive indices for any $S \in \mathcal{S}$. We will first prove the bound for any valid parsing set, and then define the optimal alignment parsing set that would lead to the tightest bound.

We first analyze a generic alignment term, using the additive decomposition into leading and remainder terms. Remember that for layer $l$ we denote the row $\ell_1$ norms with $\boldsymbol{\nu}^l = (\nu_1^l, \ldots, \nu_h^l)$, and w.l.o.g. assume that the rows are ordered in descending order according to $\nu_l$. Also note that $\|\mathbf{W}_k^l\|_\infty = \|\mathbf{W}^l\|_\infty = \nu_1^l$.

$$\|\mathbf{W}^{l+1}\mathbf{D}^l\mathbf{W}^l\|_\infty \leq \|\mathbf{W}_k^{l+1}\mathbf{D}^l\mathbf{W}_k^l\|_\infty + \|\mathbf{W}_k^{l+1}\mathbf{D}^l\mathbf{W}_r^l\|_\infty$$
$$+ \|\mathbf{W}_r^{l+1}\mathbf{D}^l\mathbf{W}_k^l\|_\infty + \|\mathbf{W}_r^{l+1}\mathbf{D}^l\mathbf{W}_r^l\|_\infty \tag{35}$$

$$\leq \|\mathbf{W}_k^{l+1}\mathbf{D}^l\mathbf{W}_k^l\|_\infty + \|\mathbf{W}_k^{l+1}\|_\infty\|\mathbf{W}_r^l\|_\infty$$
$$+ \|\mathbf{W}_r^{l+1}\|_\infty\|\mathbf{W}_k^l\|_\infty + \|\mathbf{W}_r^{l+1}\|_\infty\|\mathbf{W}_r^l\|_\infty \tag{36}$$

$$\leq \|\mathbf{W}^{l+1}\|_\infty\|\mathbf{W}^l\|_\infty\Big(\frac{\|\mathbf{W}_k^{l+1}\mathbf{D}^l\mathbf{W}_k^l\|_\infty}{\|\mathbf{W}^{l+1}\|_\infty\|\mathbf{W}^l\|_\infty} + \frac{\nu_{k+1}^l}{\nu_1^l}$$
$$+ \frac{\nu_{k+1}^{l+1}}{\nu_1^{l+1}} + \frac{\nu_{k+1}^l\nu_{k+1}^{l+1}}{\nu_1^l\nu_1^{l+1}}\Big). \tag{37}$$

$$\leq \|\mathbf{W}^{l+1}\|_\infty\|\mathbf{W}^l\|_\infty\left(\frac{\|\mathbf{W}_k^{l+1}\mathbf{D}^l\mathbf{W}_k^l\|_\infty}{\|\mathbf{W}^{l+1}\|_\infty\|\mathbf{W}^l\|_\infty} + R_\infty(\epsilon)\right). \tag{38}$$

Since the remaining, standalone layer norms also contribute $\|\mathbf{W}^l\|_\infty$, we have

$$\|\mathbf{P}(\mathbf{D})\|_\infty \leq \prod_{l=1}^{\lambda} \|\mathbf{W}^l\|_\infty \prod_{l \in S} \left( \frac{\|\mathbf{W}_k^{l+1}\mathbf{D}^l\mathbf{W}_k^l\|_\infty}{\|\mathbf{W}^{l+1}\|_\infty\|\mathbf{W}^l\|_\infty} + R_\infty(\epsilon) \right). \tag{39}$$

Bounding the Lipschitz constant accordingly:

$$L_\Phi^\infty \leq \max_{\mathbf{D}^1,\ldots,\mathbf{D}^\lambda} \prod_{l=1}^{\lambda} \|\mathbf{W}^l\|_\infty \prod_{l=1}^{\lambda-1} \left( \frac{\|\mathbf{W}_k^{l+1}\mathbf{D}^l\mathbf{W}_k^l\|_\infty}{\|\mathbf{W}^{l+1}\|_\infty\|\mathbf{W}^l\|_\infty} + R_\infty(\epsilon) \right) \tag{40}$$

$$= \prod_{l=1}^{\lambda} \|\mathbf{W}^l\|_\infty \prod_{l \in S} \left( \max_{\mathbf{D} \in \mathcal{D}} \frac{\|\mathbf{W}_k^{l+1}\mathbf{D}\mathbf{W}_k^l\|_\infty}{\|\mathbf{W}^{l+1}\|_\infty\|\mathbf{W}^l\|_\infty} + R_\infty(\epsilon) \right) \tag{41}$$

$$= \prod_{l=1}^{\lambda} \|\mathbf{W}^l\|_\infty \prod_{l \in S} A_\infty^*(\mathbf{W}^{l+1}, \mathbf{W}^l) + R_\infty(\epsilon). \tag{42}$$

Contributing an alignment term of 1 for $\{l \mid l \notin S, l+1 \notin S\}$ gives the desired result if $S = S_{opt}$, which we define below.

Given multiple valid parsing sets are possible whenever $\lambda > 2$, we lastly define the *optimal alignment parsing set*, $S_{opt}$.

**Definition A.4** (Optimal Alignment Parsing Set). *The Optimal Alignment Parsing Set $S_{opt}$ is a set in $\mathcal{S}$ that achieves the minimum product of the corresponding maximum alignment factors:*

$$S_{opt} \in \arg\min_{S \in \mathcal{S}} \prod_{l \in S} A_{\infty,l}^*. \tag{43}$$

*Note that $S_{opt}$ might not be unique, but $\min_{S \in \mathcal{S}} \prod_{l \in S} A_{\infty,l}^*$ is.*

**Complexity of finding** $S_{opt}$**:** Finding $S_{opt} \in \arg\min_{S \in \mathcal{S}} \prod_{l \in S} A_{\infty,l}^*$ is equivalent to finding the independent set $S$ in the path graph $G = (V, E)$ with $V = \{1, \ldots, L-1\}$ that maximizes $\sum_{l \in S} w_l$, where weights $w_l = -\log A_{\infty,l}^*$ (assuming $A_{\infty,l}^* > 0$; we handle $A_{\infty,l}^* = 0$ as a special case yielding $\prod_{l \in S_{opt}} A_{\infty,l}^* = 0$). This is the Maximum Weight Independent Set, which can be solved in linear time in chordal graphs, of which path graphs are a subfamily (Frank, 1976).

**(b) Spectral compressibility:** We can upper bound $L_\Phi^2$ by considering all possible activation patterns (all possible binary diagonal matrices $\mathbf{D}^l$):

$$L_\Phi^2 \leq \max_{\mathbf{D}^1,\ldots,\mathbf{D}^\lambda} \|\mathbf{P}(\mathbf{D})\|_2 \tag{44}$$

We modify the SVD decomposition for layers as

$$\mathbf{W}^l = \mathbf{U}^l\sqrt{\boldsymbol{\Sigma}^l}\sqrt{\boldsymbol{\Sigma}^l}(\mathbf{V}^l)^\top \tag{45}$$

$$= \underbrace{\left( \mathbf{U}_k^l\sqrt{\boldsymbol{\Sigma}_k^l} + \mathbf{U}_r^l\sqrt{\boldsymbol{\Sigma}_r^l} \right)}_{\mathbf{A}^l} \underbrace{\left( \sqrt{\boldsymbol{\Sigma}_k^l}(\mathbf{V}_k^l)^\top + \sqrt{\boldsymbol{\Sigma}_r^l}(\mathbf{V}_r^l)^\top \right)}_{\mathbf{B}^l}. \tag{46}$$

Note that we assume untruncated singular vector matrices for $\mathbf{W}_k^l$ and $\mathbf{W}_r^l$ for the equation above to be valid. We then decompose the spectral norm using the submultiplicative property:

$$\|\mathbf{P}(\mathbf{D})\|_2 = \|\mathbf{D}^\lambda\mathbf{W}^\lambda\mathbf{D}^{\lambda-1}\mathbf{W}^{\lambda-1}\mathbf{D}^{\lambda-2}\ldots\mathbf{D}^1\mathbf{W}^1\|_2 \tag{47}$$

$$\leq \|\mathbf{A}^\lambda\|_2\|\mathbf{B}^\lambda\mathbf{D}^{\lambda-1}\mathbf{A}^{\lambda-1}\|_2\|\mathbf{B}^{\lambda-1}\mathbf{D}^{\lambda-2}\mathbf{A}^{\lambda-2}\|_2$$
$$\ldots\|\mathbf{B}^{l+1}\mathbf{D}^l\mathbf{A}^l\|_2\ldots\|\mathbf{B}^2\mathbf{D}^1\mathbf{A}^1\|_2\|\mathbf{B}^1\|_2 \tag{48}$$

We then analyze the central term $\|\mathbf{B}^{l+1}\mathbf{D}^l\mathbf{A}^l\|_2$, and decompose it using the submultiplicative and subadditivity properties. Remember that for layer $l$ we denote the singular values with $\boldsymbol{\sigma}^l =$

710    $(\sigma_1^l, \ldots, \sigma_h^l)$. Also note that $\|\mathbf{W}_k^l\|_2 = \|\mathbf{W}^l\|_2 = \sigma_1^l$.

$$\|\mathbf{B}^{l+1}\mathbf{D}^l\mathbf{A}^l\|_2$$

$$\leq \|\sqrt{\boldsymbol{\Sigma}_k^{l+1}}(\mathbf{V}_k^{l+1})^\top\mathbf{D}^l\mathbf{U}_k^l\sqrt{\boldsymbol{\Sigma}_k^l}\|_2 + \|\sqrt{\boldsymbol{\Sigma}_k^{l+1}}(\mathbf{V}_k^{l+1})^\top\mathbf{D}^l\mathbf{U}_r^l\sqrt{\boldsymbol{\Sigma}_r^l}\|_2$$

$$+ \|\sqrt{\boldsymbol{\Sigma}_r^{l+1}}(\mathbf{V}_r^{l+1})^\top\mathbf{D}^l\mathbf{U}_k^l\sqrt{\boldsymbol{\Sigma}_k^l}\|_2 + \|\sqrt{\boldsymbol{\Sigma}_r^{l+1}}(\mathbf{V}_r^{l+1})^\top\mathbf{D}^l\mathbf{U}_r^l\sqrt{\boldsymbol{\Sigma}_r^l}\|_2 \tag{49}$$

$$\leq \|\sqrt{\boldsymbol{\Sigma}_k^{l+1}}(\mathbf{V}_k^{l+1})^\top\mathbf{D}^l\mathbf{U}_k^l\sqrt{\boldsymbol{\Sigma}_k^l}\|_2 + \sqrt{\sigma_1^{l+1}}\|(\mathbf{V}_k^{l+1})^\top\mathbf{D}^l\mathbf{U}_r^l\|_2\sqrt{\sigma_{k+1}^l}$$

$$+ \sqrt{\sigma_{k+1}^{l+1}}\|(\mathbf{V}_r^{l+1})^\top\mathbf{D}^l\mathbf{U}_r^l\|_2\sqrt{\sigma_1^l} + \sqrt{\sigma_{k+1}^{l+1}}\|(\mathbf{V}_r^{l+1})^\top\mathbf{D}^l\mathbf{U}_r^l\|_2\sqrt{\sigma_{k+1}^l} \tag{50}$$

$$\leq \sqrt{\sigma_1^{l+1}}\sqrt{\sigma_1^l}\left(\frac{\|\sqrt{\boldsymbol{\Sigma}_k^{l+1}}(\mathbf{V}_k^{l+1})^\top\mathbf{D}^l\mathbf{U}_k^l\sqrt{\boldsymbol{\Sigma}_k^l}\|_2}{\sqrt{\sigma_1^l\sigma_1^{l+1}}} + \sqrt{\frac{\sigma_{k+1}^l}{\sigma_1^l}} + \sqrt{\frac{\sigma_{k+1}^{l+1}}{\sigma_1^{l+1}}} + \sqrt{\frac{\sigma_{k+1}^l\sigma_{k+1}^{l+1}}{\sigma_1^l\sigma_1^{l+1}}}\right)$$

$$\tag{51}$$

$$\leq \sqrt{\sigma_1^{l+1}}\sqrt{\sigma_1^l}\left(\frac{\|\sqrt{\boldsymbol{\Sigma}_k^{l+1}}(\mathbf{V}_k^{l+1})^\top\mathbf{D}^l\mathbf{U}_k^l\sqrt{\boldsymbol{\Sigma}_k^l}\|_2}{\sqrt{\sigma_1^l\sigma_1^{l+1}}} + R_2(\epsilon)\right), \tag{52}$$

711    where we set all cross-alignment terms other than dominant-dominant interaction to 1. This is made
712    possible by the fact that they are the multiplication of orthogonal matrices and a ReLU matrix, all
713    of which have spectral norms upper bounded by 1. Note that for all layers $l \in 1, \ldots, \lambda$, $\sqrt{\sigma_1^l}$ will
714    appear twice in the multiplication, including the first and last layers due to the leading and final terms
715    in (48), leading to the expression:

$$\|\mathbf{P}(\mathbf{D})\|_2 \leq \prod_{l=1}^{\lambda}\|\mathbf{W}^l\|_2 \prod_{l=1}^{\lambda-1}\left(\frac{\|\sqrt{\boldsymbol{\Sigma}_k^{l+1}}(\mathbf{V}_k^{l+1})^\top\mathbf{D}^l\mathbf{U}_k^l\sqrt{\boldsymbol{\Sigma}_k^l}\|_2}{\sqrt{\sigma_1^l\sigma_1^{l+1}}} + R_2(\epsilon)\right) \tag{53}$$

716    Bounding the Lipschitz constant:

$$L_\Phi^2 \leq \max_{\mathbf{D}^1,\ldots,\mathbf{D}^\lambda}\|\mathbf{P}(\mathbf{D})\|_2 \tag{54}$$

$$\leq \max_{\mathbf{D}^1,\ldots,\mathbf{D}^\lambda}\prod_{l=1}^{\lambda}\|\mathbf{W}^l\|_2 \prod_{l=1}^{\lambda-1}\left(\frac{\|\sqrt{\boldsymbol{\Sigma}_k^{l+1}}(\mathbf{V}_k^{l+1})^\top\mathbf{D}^l\mathbf{U}_k^l\sqrt{\boldsymbol{\Sigma}_k^l}\|_2}{\sqrt{\sigma_1^l\sigma_1^{l+1}}} + R_2(\epsilon)\right) \tag{55}$$

$$\leq \prod_{l=1}^{\lambda}\|\mathbf{W}^l\|_2 \prod_{l=1}^{\lambda-1}\left(\max_{\mathbf{D}\in\mathcal{D}}\frac{\|\sqrt{\boldsymbol{\Sigma}_k^{l+1}}(\mathbf{V}_k^{l+1})^\top\mathbf{D}^l\mathbf{U}_k^l\sqrt{\boldsymbol{\Sigma}_k^l}\|_2}{\sqrt{\sigma_1^l\sigma_1^{l+1}}} + R_2(\epsilon)\right) \tag{56}$$

$$\leq \prod_{l=1}^{\lambda}\|\mathbf{W}^l\|_2 \prod_{l=1}^{\lambda-1} A_2^*(\mathbf{W}_k^{l+1}, \mathbf{W}_k^l), \tag{57}$$

717    yielding the desired result.

718                                                                        $\Box$

719    *Proof of Corollary 3.3.* Let $\boldsymbol{a}$ denote the adversarial perturbation on the input $\boldsymbol{x}$, where $\|\boldsymbol{a}\|_p \leq \delta$.
720    We define the *effective perturbation budget* in $\ell_p$ norm for the feature encoder $\Phi_k$ as $\delta_p^{\Phi_k} :=$
721    $\max\|\Phi(x) - \Phi(x+p)\|_p$. Note that by definition of the Lipschitz constant and by Thm 3.2, we have

$$\delta_p^\Phi = \max\|\Phi(\boldsymbol{x}) - \Phi(\boldsymbol{x}+\boldsymbol{a})\|_p \leq \|\boldsymbol{x} - (\boldsymbol{x}+\boldsymbol{a})\|_p L_\Phi^2 \leq \|\boldsymbol{a}\|_p \tilde{L}_\Phi^2 = \delta\tilde{L}_\Phi^2. \tag{58}$$

722    Plugging the result back into Eq (13) yields the desired result.                       $\Box$

723    **Lemma A.5.** *Under the conditions described in Thm 3.2, $R_p(\epsilon) \to 0$ as $\epsilon \to 0$ for $p \in \{2, \infty\}$.*

*Proof.* $p = \infty$: Due to the definition of compressibility, for all $l \in [\lambda]$,

$$\|\boldsymbol{\nu}^l - \boldsymbol{\nu}_k^l\|_1 \leq \epsilon \|\boldsymbol{\nu}^l\|_1 \tag{59}$$

$$\nu_{k+1}^l \leq \epsilon h \|\mathbf{W}^l\|_F, \tag{60}$$

by applying standard norm inequalities across rows and columns. The result follows from noting that the final inequality applies to both $\nu_{k+1}^l$ and $\nu_{k+1}^{l+1}$.

$p = 2$: Similarly, due to the definition of compressibility, for all $l \in [\lambda]$,

$$\|\boldsymbol{\sigma}^l - \boldsymbol{\sigma}_k^l\|_1 \leq \epsilon \|\boldsymbol{\sigma}^l\|_1 \tag{61}$$

$$\sigma_{k+1}^l \leq \epsilon \sqrt{h} \|\mathbf{W}^l\|_F, \tag{62}$$

since $\|\boldsymbol{\sigma}^l\|_2 = \|W^l\|_F$. The result follows from noting that the final inequality applies to both $\sigma_{k+1}^l$ and $\sigma_{k+1}^{l+1}$. $\qquad\square$

**Lemma A.6.** *Under the conditions described in Thm 3.2, $A_p^*(\mathbf{W}^{l+1}, \mathbf{W}^l) \leq 1$ for $p \in \{2, \infty\}$.*

*Proof.* For $p = \infty$,

$$A_\infty^*(\mathbf{W}^{l+1}, \mathbf{W}^l) = \max_{\mathbf{D} \in \mathcal{D}} \frac{\|\mathbf{W}^{l+1}\mathbf{D}\mathbf{W}^l\|_\infty}{\|\mathbf{W}^{l+1}\|_\infty \|\mathbf{W}^l\|_\infty} \tag{63}$$

$$\leq \frac{\|\mathbf{W}^{l+1}\|_\infty \max_{\mathbf{D} \in \mathcal{D}} \|\mathbf{D}\|_\infty \|\mathbf{W}^l\|_\infty}{\|\mathbf{W}^{l+1}\|_\infty \|\mathbf{W}^l\|_\infty} \tag{64}$$

$$\leq \frac{\|\mathbf{W}^{l+1}\|_\infty \|\mathbf{W}^l\|_\infty}{\|\mathbf{W}^{l+1}\|_\infty \|\mathbf{W}^l\|_\infty} = 1. \tag{65}$$

The proof follows identically for $p = 2$. $\qquad\square$

# B  Additional Technical Results and Analyses

## B.1  $(q, k, \epsilon)$ compressibility vs. other notions of approximate sparsity

Our notion of $(q, k, \epsilon)$ compressibility is similar to notions exploited in machine learning previously (Amini et al., 2011; Gribonval et al., 2012; Barsbey et al., 2021; Wan et al., 2024). More specifically, when $k \ll d$ and $\epsilon \ll 1$, Definition 2.1 is equivalent to Gribonval et al. (2012)'s definition of *compressible vector*. Inspired by desiderata from an ideal metric of sparsity in the economics literature, Diao et al. (2023) recently introduced another scale-invariant notion of approximate sparsity:

**Definition B.1** (PQ Index Diao et al. (2023)). *For any $0 < p < q$, the PQ Index of a non-zero vector $\mathbf{w} \in \mathbb{R}^d$ is*

$$I_{p,q}(\mathbf{w}) = 1 - d^{\frac{1}{q} - \frac{1}{p}} \frac{\|\mathbf{w}\|_p}{\|\mathbf{w}\|_q}. \tag{66}$$

Interestingly, it is possible to directly relate this notion of sparsity to $(q, k, \epsilon)$ compressibility, as shown in the following proposition.

**Proposition B.2.** *Given $0 < p < q$, for a vector $\boldsymbol{\theta}$, its $(q, k, \epsilon)$ compressibility implies the following lower bound for its PQ Index:*

$$1 - \epsilon - \kappa^\phi \leq I_{p,q}(\boldsymbol{\theta}), \tag{67}$$

*where $\kappa = k/d$ and $\phi = \frac{1}{p} - \frac{1}{q}$. Note that the constraints on $p, q$ imply $\phi > 0$.*

*Proof.* Let $\gamma = \frac{1}{p} - \frac{1}{q}$. Note that from (12) we know that $\|\boldsymbol{\theta}\|_p \leq (k^\gamma + d^\gamma \epsilon) \|\boldsymbol{\theta}\|_q$. This implies

$$\frac{\|\boldsymbol{\theta}\|_p}{\|\boldsymbol{\theta}\|_q} \leq k^\gamma + d^\gamma \epsilon. \tag{68}$$

Note that PQ Index from (66) can be written as $(1 - I_{p,q}(\boldsymbol{\theta}))d^\gamma = \frac{\|\boldsymbol{\theta}\|_p}{\|\boldsymbol{\theta}\|_q}$. Plugging this into the LHS of (68) and simple algebraic manipulation gives the desired result. $\qquad\square$

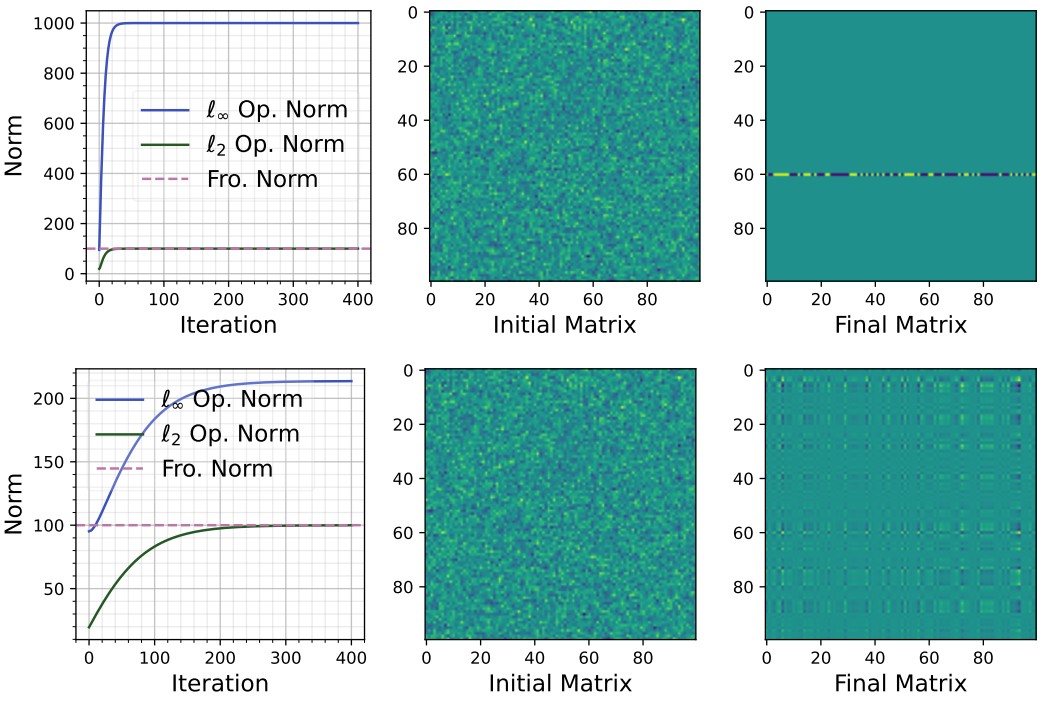

Figure 7: Optimizing for $\ell_\infty$ (top) and $\ell_2$ (bottom) operator norms.

*Remark* B.3. Assume that $\boldsymbol{\theta}$ and $\boldsymbol{\theta}'$ are $(q, k, \epsilon)$ and $(q, k', \epsilon')$ compressible respectively. If $k = k'$ and $\epsilon < \epsilon'$; or $k < k'$ and $\epsilon = \epsilon'$ implies a larger lower bound on PQI. That is, a larger $(q, k, \epsilon)$ compressibility suggests a larger PQI.

## B.2 Relationships between operator norms

Although Thm 3.1 directly relates $\ell_\infty$ and $\ell_2$ operator norms to neuron and spectral compressibility, both the known norm inequality relationships and our results on cross-norm adversarial attacks imply that these two quantities are likely to be strongly correlated under this context. We conduct simple experiment to test this hypothesis: We optimize for either $\ell_\infty$ or $\ell_2$ operator norm of a random i.i.d. Gaussian matrix $\mathbf{A}$ where $A_{i,j} \overset{\text{i.i.d.}}{\sim} \mathcal{N}(0, 1)$. We then conduct a gradient ascent-based optimization of the matrix's either $\ell_\infty$ or $\ell_2$ operator norms, while normalizing the Frobenius norm to its initialization value. In Fig. 7, as an average of 10 random seeds, we show how $\ell_\infty$ and $\ell_2$ evolve while either $\ell_\infty$ (top) and $\ell_2$ (bottom) are optimized. We note that in both case both norms are strongly associated in increasing simultaneously. Note that given the inequality $\|\mathbf{A}\|_2 \leq \|\mathbf{A}\|_F$, by the end of optimization the spectral norm reaches its limit in Frobenius norm. While the left column shows the norms across iterations, center and right columns portray the qualitative differences produced by optimizing for either columns. As expected, optimizing for $\ell_\infty$ collects all energy in a single row, while optimizing for $\ell_2$ produces a 1-rank matrix.

## B.3 Empirical analyses of the robustness bound and related quantities

In this section, we directly investigate how well our bound correlates with the adversarial robustness gap, as predicted in Corollary 3.3. In order to fully conform to the setting of Corollary 3.3, we convert the previously introduced MNIST dataset to a binary classification task by converting its labels to 0-1, by assigning 0-4 to class 0 and 5-9 to class 1. We create a fully connected network (FCN) with two hidden layers of width 300, with ReLU activations after each layer. We then create networks with various spectral compressibility through varying the rank of the hidden layers, imposed through low-rank factorization. While computing the bound, we determine $k$ (num. dominant terms), and compute $\epsilon$ and $\beta$ as statistics. Note that if $\beta = 1$, this would make the bound undefined - however, instead of being a numerical problem, this implies that $k$ should be selected lower, as dominant terms

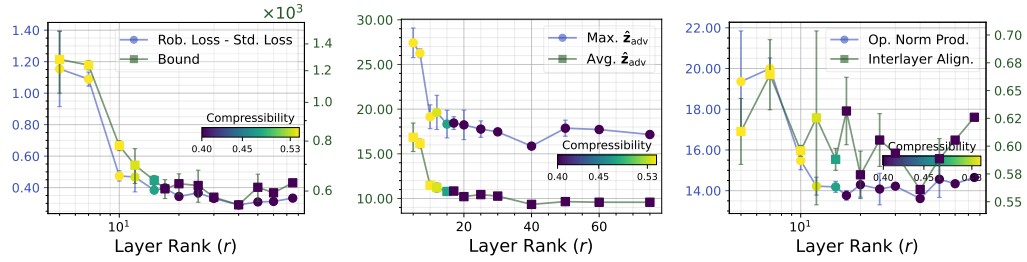

Figure 8: Empirically investigating the implications of Thm 3.2.

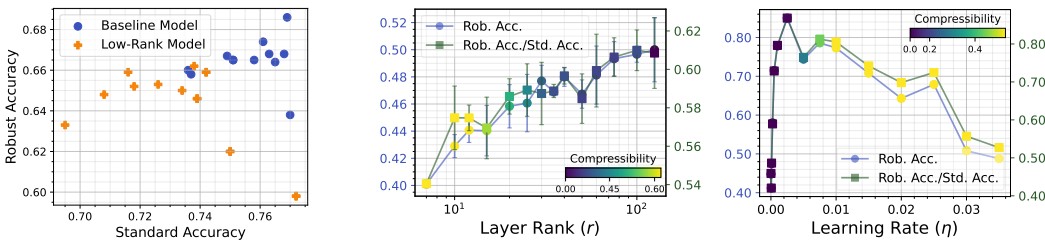

Figure 9: Adversarial fine-tuning (left) and training (center). Robust accuracy under increasing learning rate (right).

including 0 is an undesired corner case. Fig. 8 demonstrates the results of our experiment. First, Fig. 8 (left) shows that our bound is closely correlated with adversarial robustness gap. This shows that although our bound is an order of magnitude above the empirical loss difference, it is still a faithful indicator of adversarial robustness.

We then investigate whether local input sensitivity of the network tracks its global properties. As in the main paper, letting $z = \Phi(x)$ and $z_{\mathrm{adv}} = \Phi(x + a^*)$ denote the learned representations of clean and perturbed input images, we compute $\|z - z_{\mathrm{adv}}\|_2/\|a^*\|_2$ for 1000 test samples. We take this metric as a secant approximation of the local Lipschitz constant around input $x$. We then use the maximum and the mean of this statistic over the samples as *empirical lower bounds* to the global and expected local Lipschitz constants respectively. Fig. 8 (center) shows that these two values are closely correlated: An increase in the maximum sensitivity to perturbation is reflected in a similar increase in the average sensitivity. Lastly, Fig. 8 (right) investigates the effect of spectral compressibility on interlayer alignment, in parallel to product of spectral norms of the layers (to quantify the intra- vs. interlayer dynamics in our bound). Results show that while norms increase as expected, interlayer alignment does not necessarily portray a consistent pattern. We consider how and why interlayer alignment changes in response to various compressibility inducing sparsity and training dynamics to be a crucial future research direction.

## B.4 Approximating the interlayer alignment terms

Note that the interlayer alignment terms used in Thm 3.2 lead to a combinatorial optimization problem due to the discreteness of ReLU gradients, i.e. $\{0, 1\}$. A closely related precedent from the literature is SeqLip by Scaman & Virmaux (2018), with the differences relating to the normalization of the terms, and the $k$-term adaptation. However, since these differences do not lead to any changes with respect to the optimization of these terms (*i.e.* their maxima), the authors' approximation methodology is an attractive choice for determining $A_p^*$. Scaman & Virmaux (2018) report that their gradient-ascent based greedy search algorithm is in $\sim 1\%$ of the analytical solution for cases where the latter is computationally feasible. We adopt their solution to our case for both interlayer alignment terms.

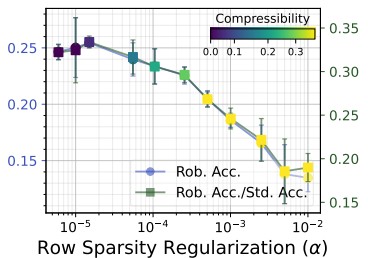

Figure 10: Effects of standard group lasso on compressibility and adversarial robustness.

# C  Details of the Experimental Setting

**Datasets.** Our experiments are conducted using the most commonly utilized datasets and architectures in research on adversarial robustness under pruning (Piras et al., 2024). Our datasets include MNIST (Deng, 2012), CIFAR-10, CIFAR-100 (Krizhevsky & Hinton, 2009), SVHN (Netzer et al., 2011). As detailed in Appendix B, we convert MNIST into a binary classification task for empirically investigating how our bound correlates with adversarial robustness gap. In all datasets, we use the canonical train, test splits. Whenever validation set-based model selection or early stopping is used, we utilize $5\%$ of the training set for this task, and conduct early stopping with a patience of 10 epochs based on validation loss.

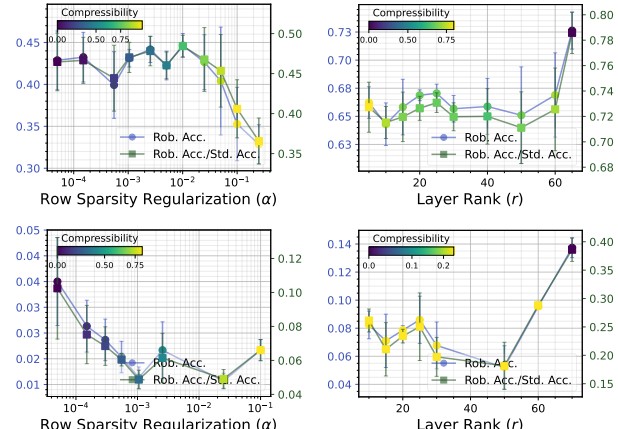

Figure 11: Results with SVHN & Wide ResNet 101-2 (top), CIFAR-100 & VGG16 (bottom).

**Models.** Architectures we utilize include fully connected networks (FCN), ResNet18 (He et al., 2016), VGG16 (Simonyan & Zisserman, 2014), and WideResNet-101-2 (Zagoruyko & Komodakis, 2016). Whenever needed, we apply modifications to the standard architectures in question. For our visualization experiments at the beginning of Sec. 4, we utilize a 1-hidden layer FCN with ReLU activation, no bias nodes, and with a width of 400. For our main results with CIFAR-10, we utilize a 2000-width FCN with 4 hidden layers, with the remaining architectural choices identical. Regarding the VGG16 architecture, due to our datasets being size $32 \times 32$, we remove the redundant 4096-width linear layers (along with their interleaving dropout and ReLU layers). Lastly, when conducting the low-rank factorization experiments, we modify linear layers with a factorized layer, and do the equivalent for 2D convolutional layers (Zhong et al., 2023).

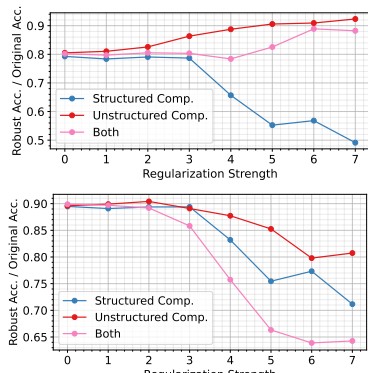

Figure 12: Unstructured alongside structured comp., for row sparsity (top) and spectral comp. (bottom).

**Standard training**. We normally use softmax cross-entropy loss, the AdamW optimizer with a weight decay of $0.01$, a learning rate of $0.001$, and use validation set based model selection for early stopping. For adversarial training tasks, we also include a cosine learning rate annealing schedule (epochs = 60, min. learning rate = 0), basic data augmentation in the form of random cropping and horizontal flips, and an adversarial validation set.

**Evaluating and training for adversarial robustness**. For evaluating adversarial robustness, we primarily employ the AutoPGD attack (Croce & Hein, 2020), using the implementation from Nicolae et al. (2018). During adversarial training, we generate adversarial examples at each iteration using the PGD attack (Madry et al., 2018). Unless stated otherwise, adversarial examples make up $50\%$ of each training minibatch. For models trained end-to-end with adversarial robustness, we set $\epsilon = 8/255$ for $\ell_\infty$ attacks and $\epsilon = 0.5$ for $\ell_2$ attacks. For standard or adversarially fine-tuned models, we use $25\%$ of these budgets to enable a clear comparison.

**Implementation**. We utilize the Python programming language and PyTorch deep learning framework for our implementation (Paszke et al., 2019). Whenever possible, we utilize the default torchvision (maintainers & contributors, 2016) implementations of our models - we modify these baselines for the changes mentioned above. For adversarial training and evaluation, we use the Adversarial Robustness Toolbox (Nicolae et al., 2018). The attached source code provides further details regarding our implementation, and will be made publicly available upon the paper's publication.

 **Hardware and resources**. All experiments are conducted on the computational server of an insitute,
utilizing Nvidia L40S GPUs. The main paper experiments took a total of 600 GPU hours to complete,
including $\geq 3$ seed replication for the main results. Total development time is estimated to be $3.5\times$
of the compute time for the final publication.

# D  Additional Empirical Results

## D.1  Experiments with other datasets and architectures

As mentioned in the main paper, we now extend our empirical findings to other datasets and architectures. Fig. 11 demonstrates results with SVHN dataset and Wide ResNet 101-2 architecture (top),
and CIFAR-100 dataset and VGG16 architecture (bottom). Our results replicate with novel datasets
and architectures, as qualitatively identical results are obtained in these alternative settings.

## D.2  Group sparsity regularization

In the main paper, we highlight that we utilize a scale-invariant version of group lasso to disentangle
the downstream effects of increasing compressibility vs. decreasing overall parameter scale. Fig. 10
replicates our main results on ResNet18 and CIFAR-10 while using standard group lasso regularization. While its effects are mostly similar to our version of group lasso, we note that Fig. 10 presents
a subtle difference, where group lasso first creates a slight but statistically significant (error bars =
1 std. deviation) increase in robustness at very low levels. However, as indicated in the main text,
these benefits are overtaken by the negative effects of row compressibility as regularization strength
increases.

## D.3  Adversarial training results for spectral compressibility

Fig. 9 (left, center) presents the spectral
compressibility counterpart for adversarial fine-tuning and training results from
the main paper, under $\ell_2$ adversarial attacks. The patterns clearly mirror those
presented in the main paper under row
sparsity conditions.

## D.4  Compressibility
## through inductive bias

We now examine whether the results we
have observed with explicit regularization methods also apply to cases when
compressibility is obtained through the
inductive bias of the learning algorithm.

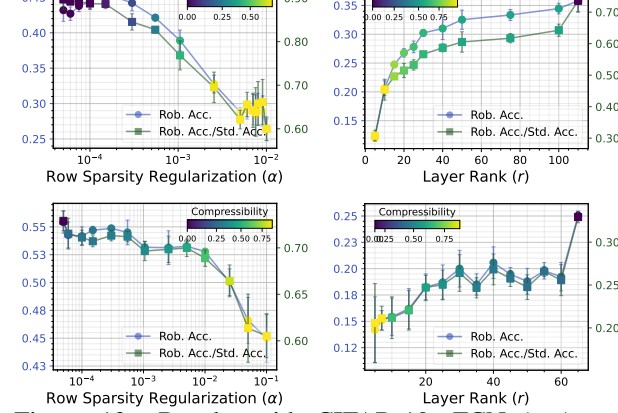

Figure 13: Results with CIFAR-10, FCN (top) and
ResNet18 (bottom), with alternative attack norms to Fig. 3

For this, we go back to the setting presented in Appendix B, and instead of increasing regularization
hyperparameter, we increase initial learning rate ($\eta$) of the training algorithm. The results, presented
Fig. 9 (right), paint an intriguing picture. While initially increasing $\eta$ *improves* adversarial robustness
under $\ell_\infty$ attacks (perhaps paralleling its well-known benefits for standard generalization), as soon as
it starts to increase row compressibility, its benefits of $\eta$ quickly disappear. This highlights the fact
that our results not only inform the adversarial robustness behavior under explicit regularization and
architecture design, but also inductive biases of the learning algorithm.

## D.5  Unstructured compressibility

While unstructured compressibility is not the focus of our study, we note that it appears in the
bound for $L_\Phi^\infty$ in Thm 3.2, unlike that for $L_\Phi^2$. To investigate the significance of this result, we
replicate the setting presented in Appendix B, but this time in addition to increasing the group
lasso/nuclear norm regularization, we run a separate set of experiments where we either solely
increase L1 regularization, or increase it along with structured sparsity-inducing regularization.

We then compare the performance of the resulting models under the corresponding adversarial attacks. The results are presented in Fig. 12. Remember that our bound implies *positive* effects of unstructured compressibility for $L_\Phi^\infty$. Indeed, in Fig. 12 we see that L1 regularization can compensate for the negative effects of structured compressibility (top), while it has no such benefits for spectral compressibility (bottom). We believe that understanding the intricate relationships among different types of compressibility is a crucial future research direction.

## D.6 Results with alternative norms

While for brevity we presented our main results to include robustness against $\ell_\infty$ attacks under neuron sparsity, and $\ell_2$ attacks under spectral compressibility, for completeness we provide our central results with the cross-norm attacks, *i.e.* $\ell_\infty$ attacks under spectral compressibility, and $\ell_2$ attacks under neuron sparsity. The results are presented in Fig. 13, and are fully in line with the results presented in the main paper.

