# OpenReview forum: "On the Interaction of Compressibility and Adversarial Robustness"
_NeurIPS.cc/2025/Workshop/Reliable_ML — NeurIPS 2025 - Reliable ML Workshop_

### Official Review · Reviewer_vbo7 · 2025-09-08
**This paper studies the relationship between compressibility and robustness. Two types of compressibility can make models more vulnerable. Theoretical results are verified by experiments and also guided solutions that can address vulnerability.**

**Rating:** 10
**Confidence:** 2

**Review:**

## Summary
This paper studies the relationship between compressibility and robustness. Two types of compressibility can make models more vulnerable. Theoretical results are verified by experiments and also guided solutions that can address vulnerability.

## Strengths
- The paper provides a novel connection between compressibility and robustness using rigorous proofs with high theoretical quality and validates them with through experiments along with mitigating strategies.

## Weaknesses / Limitations
- Some limitations were mentioned at the end of the paper.
- Defenses for compressed models can be further investigated.

## Suggestions for Authors
- Line 76: What is $S$?
- Line 81: Distribution $\pi$ is jointly on $(x, y)$. Maybe you missed $y$?
- Line 170: From the definition of spread it seems like $\beta$ depends on $k$. What is $\beta_v$ exactly? I assume that $k$ is $k_v$?
- You bound various norms by the F norm but is the F norm affected by compressibility parameters? From your experiments it seems like you keep this fixed.
- Line 210: Typo $l_2$
- Line 213: Is $\hat{y} = x^\top \theta$? It’s not the multilayer neural network output in (2).
- Line 219: What is the decomposition and the interpretation?

---

### Official Review · Reviewer_PncN · 2025-09-20
**Connection between Robustness and Compression via the Effect on Norms**

**Rating:** 9
**Confidence:** 3

**Review:**

This paper studies the relationship between neurons and spectral compressibility to adversarial robustness by investigating how l_2 and l_infinity norms are affected under compression. The writing is very clear, and someone from outside the field (no experience with compression and robustness), the theorems and their explanations were easy to understand.

1. For me, the most interesting thing is the definition of (q,k,eps) compressibility and Thm 3.1, because once that is established, one can predict how it can be related to the norms and then adversarial robustness. Even though Thm 3.1 are upper bounds, the effect of compressibility is immediately clear.
2.  The notation is a bit hard to parse for the interlayer alignment bounds. But similar to Theorem 3.1, it is clear how the smoothness constant can become much larger than expected under compressibility.
3.  The experiments demonstrate the central hypothesis, but I was unable to understand how (q,k,eps) compressibility is ensured. This also brings to me to the point that in practice, satisfying (q,k,eps) compressibility exactly will be tough. If the authors can provide some roadmap or results about "probabilistic" (q,k,eps), it will be interesting.

Overall, the work is very interesting and the theorems are very helpful.